# Modeling Variability in Seismic Analysis of Concrete Gravity Dams: A Parametric Analysis of Koyna and Pine Flat Dams

Bikram Kesharee Patra [1,*], Rocio L. Segura [2] and Ashutosh Bagchi [1]

1   Department of Building, Civil & Environmental Engineering, Concordia University, Montreal, QC H3G 1M8, Canada; ashutosh.bagchi@concordia.ca
2   Department of Civil, Environmental, and Sustainable Engineering, Santa Clara University, Santa Clara, CA 95053, USA; rsegura@scu.edu
*   Correspondence: b_patra@live.concordia.ca

**Abstract:** This study addresses the vital issue of the variability associated with modeling decisions in dam seismic analysis. Traditionally, structural modeling and simulations employ a progressive approach, where more complex models are gradually incorporated. For example, if previous levels indicate insufficient seismic safety margins, a more advanced analysis is then undertaken. Recognizing the constraints and evaluating the influence of various methods is essential for improving the comprehension and effectiveness of dam safety assessments. To this end, an extensive parametric study is carried out to evaluate the seismic response variability of the Koyna and Pine Flat dams using various solution approaches and model complexities. Numerical simulations are conducted in a 2D framework across three software programs, encompassing different dam system configurations. Additional complexity is introduced by simulating reservoir dynamics with Westergaard-added mass or acoustic elements. Linear and nonlinear analyses are performed, incorporating pertinent material properties, employing the concrete damage plasticity model in the latter. Modal parameters and crest displacement time histories are used to highlight variability among the selected solution procedures and model complexities. Finally, recommendations are made regarding the adequacy and robustness of each method, specifying the scenarios in which they are most effectively applied.

**Keywords:** dam seismic assessment; uncertainty analysis; model complexity; fluid–structure–soil interaction; nonlinear analysis; numerical simulation

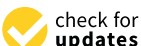



## 1. Introduction

Seismic assessment plays a crucial role in ensuring the safety and performance of critical infrastructure such as dams. It involves the evaluation of the structural response of these systems under extreme events, such as seismic loading and compliance with regulatory guidelines. However, this assessment is subjected to various degrees of uncertainties arising from the inherent complexity of the system, the lack of knowledge about the seismic hazard, and the modeling assumptions made during the analysis. These uncertainties can significantly impact the behavior of the dam system and related decisions, highlighting the need for a comprehensive understanding of their effects.

Uncertainties influencing seismic assessments can be categorized as aleatory and epistemic. Epistemic uncertainty stems from limited knowledge and arises due to data gaps, incomplete comprehension of phenomena, and simplifications in modeling. It encompasses variations in model parameters, structural properties, and solution methods impacting the calculation of structural responses. In contrast, aleatory uncertainty, known as inherent or irreducible, originates from natural randomness and variability, spanning ground motion characteristics and complex systems. Unlike epistemic uncertainty, which results from insufficient information, aleatory uncertainty is intrinsic and cannot be eradicated through data or understanding enhancement. In order to improve the accuracy and robustness of

seismic assessments, it is essential to investigate and quantify the uncertainties associated with different modeling decisions. This includes the selection of solution procedures, model complexities, and the consideration of various loading and boundary conditions [1,2]. Figure 1 shows the taxonomy of uncertainty in risk analysis.

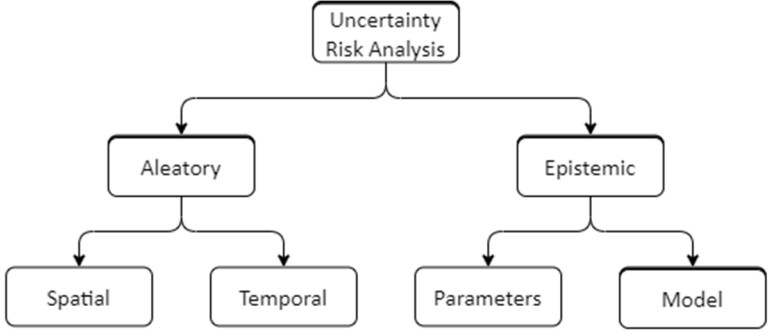

**Figure 1.** Taxonomy of uncertainty in risk analysis adapted from [1].

These uncertainties arise from various factors, such as the selection of solution procedures, model complexities, and consideration of loading and boundary conditions. By exploring and understanding these uncertainties, engineers and researchers can make informed decisions in rehabilitation and risk management processes for structures subjected to seismic hazards. The selection of solution procedures in seismic analysis involves choosing appropriate numerical methods and algorithms to simulate the dynamic behavior of structures. Further, different solution procedures may yield varying results due to the inherent assumptions and limitations of each method [2,3]. Understanding the uncertainties associated with different solution procedures allows for the identification of the most suitable approach for different stages of seismic assessments.

Model complexities play a crucial role in capturing the behavior of real-world structures.

It is essential to underscore the importance of a progressive approach in dam safety assessment, which involves the use of increasingly complex models. This is a particularly valuable approach in the dam industry, where intricate analyses are not always feasible for every situation. The level of detail in modeling influences the accuracy of predictions and the ability to account for important structural features [4,5]. The selection of the method for safety assessment is influenced by various factors: the structure's scale and potential damage consequences, its current state (well-maintained or deteriorated), and the required precision of the analysis [6]. Moreover, the impact of modeling uncertainty on system response parameters can be mitigated by incorporating a robust verification and validation framework [7]. In the study presented in [7], model variability arising from epistemic uncertainty was thoroughly investigated using data from the International Commission on Large Dams (ICOLD) and United States Society on Dams (USSD) benchmark studies [8,9], and the quantification of modeling variability was carried out using dispersion or logarithmic standard deviation. A comprehensive understanding of modeling uncertainty is imperative for enhancing the reliability of seismic assessments in dam safety. The chosen modeling approach must also consider the trade-off between the need for accuracy and detail with the practical constraints of time, resources, and available data, which will also influence risk management decisions. Traditionally, if simpler and more expedited analyses (such as 2D, linear, and pseudo-static approaches) do not meet the minimum seismic safety requirements, a more detailed analysis is necessary. Understanding the variability introduced by the different layers of model complexities is critical to help engineers determine the level of detail required for reliable assessments and assess the sensitivity of structural responses to different modeling choices [10]. To this end, this paper aims to evaluate a dam's response variability associated with different modeling configurations within a progressive seismic analysis framework. To achieve this, a detailed parametric study incorporating various solution approaches and model complexities will be used. The assessment will be carried out using different software such as EAGD-84, Abaqus 6.14, and ADRFS v1 [11–13]. The response of the dam only, as well as the interaction

with the foundation and reservoir, will be considered. The modeling variability in the seismic response is evaluated through the comparison of modal parameters and crest displacement time histories, across the different modeling configurations. The methodology is applied to two case study dams with well-documented geometric, material, and dynamic properties, Koyna and Pine Flat. These findings are essential for the improvement of safety assessment, shedding light on the variations associated with different modeling decisions, advantages, and disadvantages and the adequacy of each approach for different dam safety analysis stages where varying complexity methods might be required.

## 2. Seismic Analysis of Dams

The seismic analysis of concrete dams is mostly based on numerical simulations, employing appropriate analysis methodologies to reliably estimate system responses [14,15]. During the seismic safety assessment, careful consideration must be applied at every stage. This includes ensuring the use of accurate material properties, selecting appropriate ground motion, proper scaling, choosing a suitable numerical model (either 2D or 3D), applying accurate boundary conditions, and employing the right analysis techniques [16–18]. As depicted in Figure 2, the seismic performance of a concrete gravity dam must be evaluated progressively, i.e., considering increasing complexity, computing effort, and aligning with the assessment objective [6,19–21]. In this study, we employed both linear and nonlinear analyses with varying system configurations. According to Figure 2, linear analysis offers good capability with low computational effort, while nonlinear analysis provides very good capability with moderate computational effort. The following sections further describe the different analyses considered in this study.

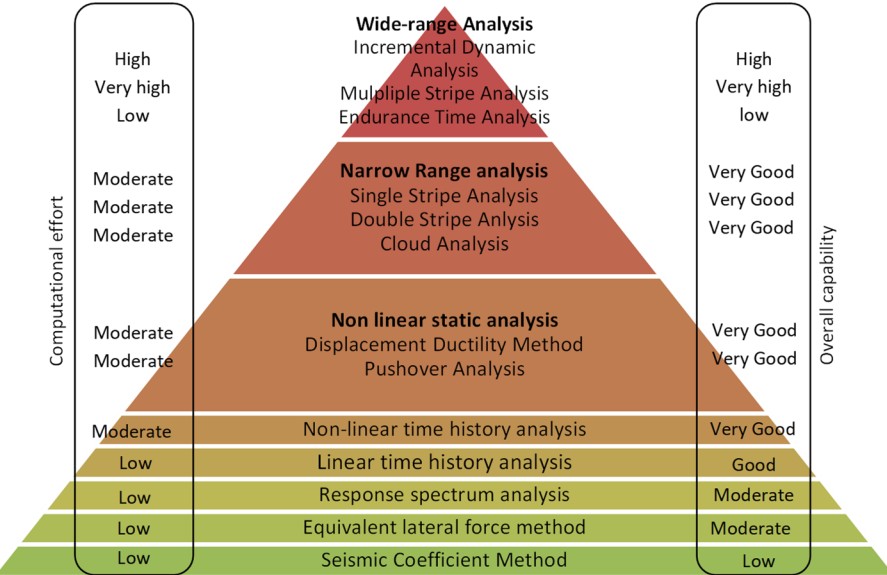

**Figure 2.** Progressive seismic safety evaluation adapted from [21].

### 2.1. Linear vs. Nonlinear Analysis

Based on the solution objective, required computational stability, and accuracy, dynamic analysis can be linear or nonlinear, and it can be either in the time or frequency domain.

A linear analysis can provide information regarding overstressed areas but for understanding the actual dynamic behavior, a nonlinear analysis is required [22,23]. The response of a structure (stress and deflection) can be estimated using linear analysis, suggesting possible deterioration. A linear elastic analysis with increased safety margins is sufficient for dams subjected to an operating or design basis earthquake (OBE or DBE). Linear studies can predict damage, but not failure since stress redistribution is caused by fractures or opening, and closing of contraction joints is not considered [19]. Furthermore, simplifying assumptions are made in linear analysis, for example, modeling monolithic

dams (ignoring contraction joints and weak lift lines), monolithic foundations (ignoring joints and discontinuities), added mass for hydrodynamic interactions, and linear elastic material models [19,24,25]. Because of these assumptions, linear assessments are fraught with ambiguity and always fail to fully explain the structure's true behavior. By removing the simplifying assumptions of linear analysis, nonlinear analysis can yield more accurate estimates of a structure's dynamic behavior [26]. As a result, nonlinear analysis can estimate, in a more realistic way, the likelihood of failure and damage levels, which are essential for seismic risk assessment.

### 2.2. Horizontal vs. Combined Horizontal and Vertical Earthquake Components

In the context of dam seismic assessment, analyzing horizontal earthquake components alone as opposed to combined horizontal and vertical components introduces different considerations for stability analysis. Both horizontal and vertical ground motions influence the seismic response of dams, each contributing differently to the overall structural behavior. When considering horizontal earthquake components alone, the focus is primarily on the lateral forces and torsional effects that dams may experience during seismic events. This approach is especially relevant for assessing the stability of dam structures against sliding, overturning, and other modes of failure caused by lateral ground motion. Horizontal ground motions are crucial for understanding the potential impact of seismic events on the dam's integrity, its foundation stability, and its interaction with the reservoir [27].

On the other hand, incorporating both horizontal and vertical components provides a more comprehensive representation of the dynamic forces acting on a dam. Vertical ground motions introduce additional effects, such as uplift and dam–water interactions, which can significantly influence the structural response and potential failure modes. Vertical motions also contribute to the potential for base sliding, foundation settlement, and changes in the reservoir water level [28].

Two-dimensional (2D) analyses of the dam geometries are still the most common approach for the design or evaluation of gravity dams. Most three-dimensional analysis methods in the past were developed in reference to arch dams [29]. In this type of analysis, a single (horizontal) or two-component (horizontal and vertical) approach is used for seismic loading. However, when incorporating the second horizontal component, the analysis becomes three-dimensional (3D), providing a more realistic representation of the complex interaction between the dam, foundation, and reservoir. The choice between analyzing with only one horizontal component, with a single horizontal and vertical component, or with both horizontal components depends on the specific characteristics of the dam, the seismic hazard scenario, and the engineering objectives. The presence of the construction joints, which is a requirement for conventional concrete dam bodies, justifies the use of 2D analyses of concrete gravity dams to some extent, based on the assumption that the monoliths behave independently during seismic events. The need for 3D modeling may arise when the accuracy of 2D models proves insufficient. This requirement could be influenced by factors such as dam typology or the presence of dams in narrow canyons [30]. For critical structures like dams, it is often recommended to perform analyses using both approaches to comprehensively evaluate the potential failure modes and ensure the safety of the structure under various seismic conditions [31].

### 2.3. System Configuration

In the seismic assessment of dams, the choice of system configuration plays a vital role in accurately predicting the structural response under seismic loads. Four primary model configurations are commonly used: (i) dam only (D), (ii) dam–foundation interaction (DF), (iii) dam–reservoir interaction (DR), and (iv) dam–foundation–reservoir (DFR) system. The progressive incorporation of these complexities allows for a more realistic representation of the dynamic behavior and interaction of the entire dam system subjected to seismic forces.

Starting with the dam-only model, the structural response of the dam is analyzed without considering interactions with its foundation or reservoir. This provides a fundamental understanding of the dam's inherent stiffness and response characteristics. As the complexity increases, the dam–foundation interaction is considered, introducing the effects of the soil–structure interaction on the dam's seismic response. Next, the dam–reservoir interaction accounts for hydrodynamic effects caused by the reservoir's water mass, such as water-induced pressures and uplift forces. The ultimate level of complexity is achieved in the coupled dam–foundation–reservoir model, which captures the combined effects of all three components. As model complexity increases, damping and time periods of vibration may change and affect the energy dissipation mechanisms. The presence of the foundation and reservoir components introduces additional modes of vibration, affecting the system's natural frequencies and mode shapes.

The boundary conditions could vary based on the model configuration. As complexity increases, the boundary conditions include the interplay of structural, geological, and hydrodynamic factors included in the model. Increasing model complexity enhances the seismic assessment by providing a holistic understanding of the entire dam system's behavior [32]. By progressively incorporating dam–foundation–reservoir interactions, informed decisions can be made, optimizing design strategies and enhancing the seismic resilience of dam structures.

## 3. Modeling Variabilities in Seismic Analysis

This study underscores how varying model complexities within the same solution procedure and differences across various solution approaches can influence the dam response and subsequent safety assessments. Modeling variabilities are explored by considering different solution procedures, i.e., time or frequency domain, linear or nonlinear analysis, and model complexity, including D, DF, DR, and DFR systems. Additional complexity is introduced by simulating the dynamic interaction of the reservoir using Westergaard-added mass and acoustic elements with non-reflecting boundary conditions. The foundation is modeled with uniform material properties and three sides fixed.

The numerical simulation is conducted using three analysis tools: EAGD-84, ADRFS v1, and Abaqus 6.14, each adopting a different solution procedure. The models are uniformly constructed across the software tools, maintaining identical material parameters, loads, mesh configurations, and boundary conditions. Additionally, each software employs a unique approach to implement damping for both the dam and foundation. Specifically, EAGD-84 utilizes hysteric damping, ADRFS v1 employs a damping ratio, and Abaqus adopts Rayleigh damping. The reservoir level is set to zero while carrying out dynamic analysis for the D and DF models. Conversely, the full reservoir load is considered in the case of DR and DFR models. This study consists of two stages: firstly, conducting modal analysis on various model configurations (D, DF, DR, and DFR) to characterize the system's dynamic behavior; and secondly, estimating the maximum crest displacement while accounting for both linear and nonlinear material properties, along with either horizontal or horizontal and vertical seismic components. Table 1 provides a summary of the scenarios modeled in each software. Each simulation will be designated as shown in Equation (1) and Figure 3.

$$S \overbrace{x}^{\text{Scenario}} . \underbrace{y}_{\substack{\text{Model} \\ \text{complexity}}} . \overbrace{z}^{\text{Software}} \tag{1}$$

where x corresponds to the scenario and can take the following values:

**Table 1.** Modeling capability of software.

| Scenario, x = | Software, z = | | | |
|---|---|---|---|---|
| | 1 | 2 | 3 | 4 |
| 1 | ✓ | ✓ | ✓ | ✓ |
| 2 | ✓ | ✗ | ✓ | ✓ |
| 3 | ✗ | ✗ | ✓ | ✓ |
| 4 | ✗ | ✗ | ✓ | ✓ |

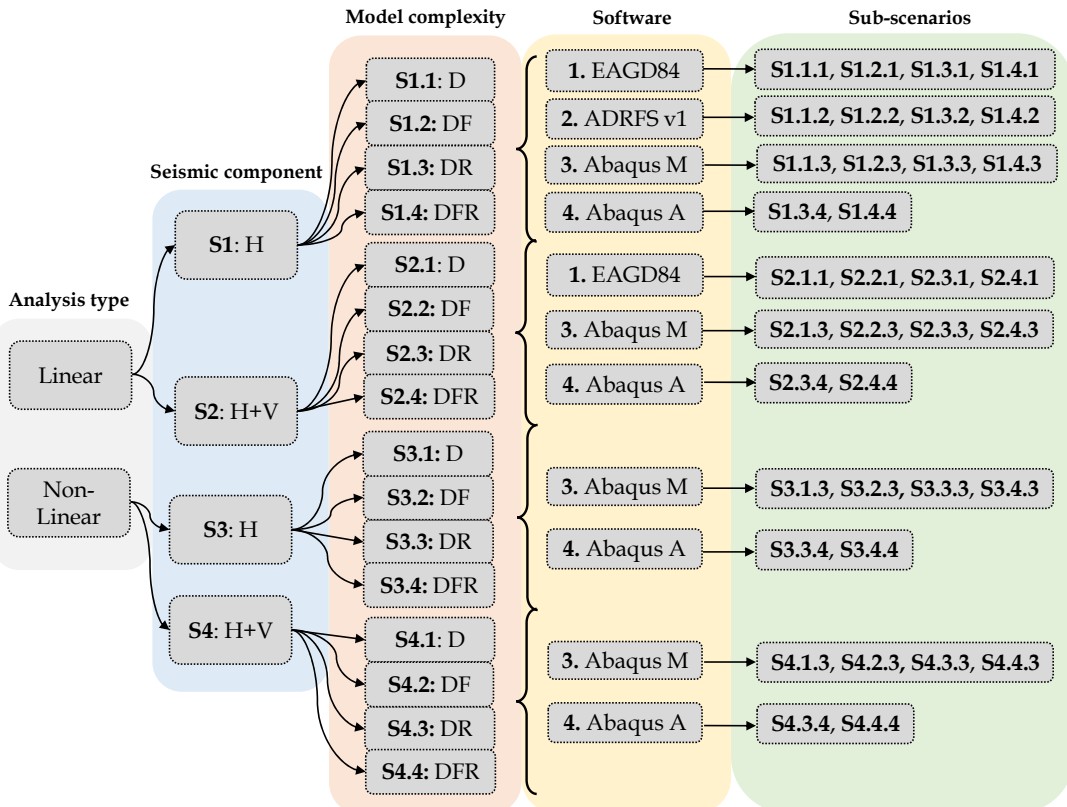

**Figure 3.** List of analysis scenarios considered for numerical simulation.

x = 1, linear analysis considering only the horizontal seismic component;
x = 2, linear analysis considering the horizontal and vertical seismic components;
x = 3, nonlinear analysis considering only the horizontal seismic component;
x = 4, nonlinear analysis considering the horizontal and vertical seismic components;
y corresponds to different model complexities configurations corresponding to:
y = 1, dam only, D;
y = 2, dam–foundation, DF;
y = 3, dam–reservoir, DR;
y = 4, dam–foundation–reservoir, DFR;
and z corresponds to the software used for that simulation:
z = 1, EAGD-84;
z = 2, ADRFS v1;
z = 3, Abaqus M (added mass);
z = 4, Abaqus A (acoustic).

The comparative analysis of different analysis scenarios using the three software tools provides insights into ease of use, flexibility or scalability for parametric studies, the time required for each analysis, and modeling capabilities. The following section discusses the chosen set of software, i.e., EAGD-84, ADRFS v1, and Abaqus 6.14, in terms of solution methodologies and modeling techniques used in this study.

### 3.1. EAGD-84

EAGD-84 software assesses the seismic response of concrete gravity dams using a substructure formulation and a frequency domain analysis approach, including dynamic dam–foundation–reservoir interactions, water compressibility, and reservoir bottom absorption [11]. The dam monolith is modeled as a 2D assemblage of planar, non-conforming four-node finite elements. The typical dam–foundation–reservoir system, as implemented in the program, is shown in Figure 4. Constant hysteretic damping represents energy dissipation in the dam concrete and foundation. The reservoir is represented as a fluid domain of constant depth and infinite length in the upstream direction, and the absorptiveness of the reservoir bottom materials is defined by a wave reflection coefficient at the reservoir bottom. To account for the effects of dam–foundation interaction, the frequency-dependent dynamic stiffness matrix for the foundation region is established with regard to the degrees of freedom of the nodal points at the dam base. Horizontal and vertical seismic components can be applied together or one at a time using free-field ground acceleration.

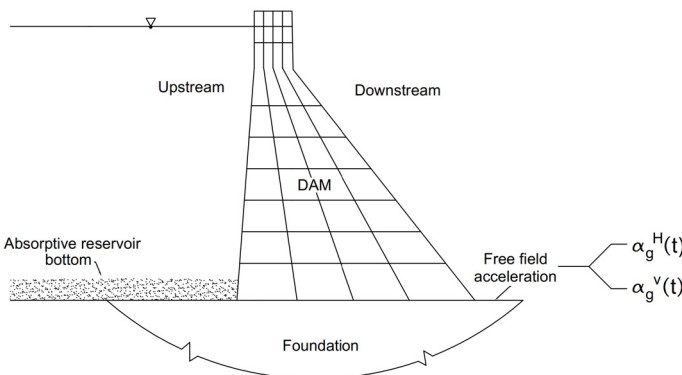

**Figure 4.** Dam–foundation–reservoir system [11].

Hydrostatic loads, nodal point displacements, and element stresses due to static loads are among the program's outputs. Modal parameters can be estimated for the dam or dam foundation with the selected interaction. Complete stress and displacement response histories for each finite element, as well as the peak maximum and minimum principal stress in each finite element, including the times of occurrence, are also provided. For both pre-processing input and post-processing output from EAGD-84, the source code can be run independently in a Windows environment or through a MATLAB 2016 [33] interface as developed by Løkke [34].

In the context of this study, the dam and foundation are set to plane stress, and linear dynamic analysis simulations with model complexities (D, DF, DR, and DFR) considering horizontal and combined horizontal and vertical components are carried out.

### 3.2. ADRFS v1

The ADRFS v1 is a GUI-based software developed on MATLAB 2016 [33] used for seismic analysis of concrete gravity dams considering dam–foundation–reservoir interactions [35]. This study uses an eight-nodded isoparametric quadrilateral, plane stress-type element for the dam and the foundation. The software accounts for the foundation flexibility and compressibility of the reservoir. The dam and foundation domains use displacement-based plane stress/strain finite element formulation, whereas, for the reservoir domain, pressure-based finite element formulation is used [35,36].

Dam and foundation damping are implemented through a damping ratio. The software considers concrete aging effects [36], reservoir bottom absorption effects [37], reservoir infiniteness using non-reflecting boundary conditions, and semi-infinite foundations using non-radiating boundary conditions. It provides solutions for individual components or a coupled system. The tool can perform modal analysis and linear time history analysis in the time domain for random vibration (as a single component of ground acceleration, either

horizontal or vertical) and harmonic excitation. The output is provided as displacement and stress history envelope values for a coupled DFR system.

The infinite computing domain of the foundation is constrained to a bounded one with truncated borders, i.e., three sides are fixed. Similarly, different truncation non-reflecting boundary conditions are implemented to consider reservoir infiniteness. However, spurious wave reflections from these truncated boundaries can propagate back into the medium's interior. To overcome the spurious wave reflections, non-reflecting boundary conditions are used to efficiently absorb the incident stress waves. In this study, the dam and foundation are modeled with eight-nodded isoperimetric quadrilateral, plain-stress elements, whereas the reservoir is represented with eight-nodded pressure-based elements. The reservoir's infiniteness is simulated using the Maity–Bhattacharya boundary condition [38,39] at the truncation boundary. The surface wave at the reservoir's free surface and the absorptive boundary at the reservoir bottom were not considered. The gravity loadings applied for the dam body and the hydrodynamic effect are considered for the scenarios with the reservoir. Linear analysis with model complexities (D, DF, DR, and DFR) was carried out only with the seismic horizontal component.

*3.3. Abaqus*

Abaqus is a general-purpose finite element-based software with an extensive library of element types and materials. It can model nearly any geometry and material model that can simulate stress and deformation in linear and nonlinear scenarios. Material nonlinearity can be introduced into Abaqus using the damaged plasticity models. In this context, a comparative analysis to identify the degree of variation between the classical concrete damage plasticity (CDP) and modified concrete damage plasticity model is carried out for the dam-only case for Koyna Dam, i.e., scenario 4.1.3.

In this study, a plane stress-type element (CPS4R) was chosen for the dam and foundation. The reservoir's hydrodynamic loading is modeled using two distinct approaches: Westergaard's added mass [37] and acoustic elements (AC2D4), hereafter referred to as "Abaqus M" and "Abaqus A", respectively. The direct coupling is implemented using tie constraints, and Rayleigh damping is used for the dam and foundation. Modal analysis for the first four modes is carried out using the Lanczos Eigen solver. The dam and foundation interfaces are tied, and the foundation bottom is fixed on three sides. When a reservoir is modeled using an acoustic element, a non-reflective boundary condition is applied at the truncation point, and a zero-pressure boundary condition is implemented at the free water surface. The dynamic interaction of DR and DF is achieved through tied constraints. Seismic loading is applied in the horizontal and vertical directions at the dam–foundation interface. Figure 5a,b show the DFR model with the reservoir modeled as added mass and acoustic, as well as other details like boundary condition and model size.

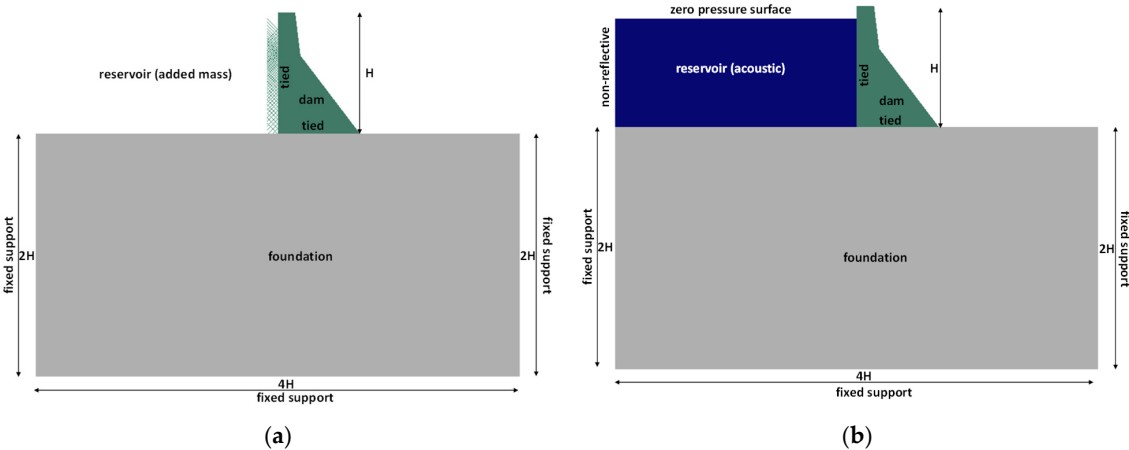

**Figure 5.** Considered Abaqus DFR models: (**a**) Abaqus M and (**b**) Abaqus A.

The modal analysis was carried out with model complexities: D, DF, DR, and DFR. Subsequently, dynamic analysis with increasing model complexity was carried out for scenarios, as shown in Figure 3.

## 4. Case Study: Koyna and Pine Flat Dams

This section provides an overview of two concrete gravity dams chosen for the case study: the Koyna and Pine Flat dams. These dams were well studied and their geometric, dynamic, and material properties are reported widely in the existing literature [22,40], making them ideal candidates for investigating the effects of diverse modeling approaches. Their selection is primarily based on common attributes such as geometric similarity, diverse seismic conditions, and the presence of comprehensive data for model creation and validation. In this study, aging effects were neglected. However, it should be noted that material properties can undergo changes over time due to aging, which impacts the seismic response. Even if aging effects are neglected, the characteristics of the case study dams are in line with our research objectives, which aim to examine dynamic response variations under distinct solution methodologies, not to estimate the current seismic behavior of a particular dam. Through this comparative analysis of the two dams, we can glean valuable insights into the determinants of their seismic responses.

### 4.1. Location and Geometry Description

Koyna Dam, located in Maharashtra, India, and Pine Flat Dam, located in California, USA, play crucial roles in water storage, hydroelectric power generation, and flood control in their respective regions. The downstream view of Koyna Dam and its cross-section, as considered in the numerical simulation, are shown in Figures 6a and 6b, respectively. Koyna Dam has a crest length of 853.5 m and a height of 85.34 m above the riverbed, reaching a depth of 103.02 m below the deepest foundation. It consists of a total of 27 monoliths, each measuring 15.24 m in width. In a similar manner, Pine Flat Dam presents a height of 130 m and a crest length of 561 m; it is comprised of 36 monoliths, each measuring 15.25 m wide, along with an additional 12.2 m wide block. Figure 7a,b illustrate the downstream view and cross-section of the dam, as considered in the numerical simulation.

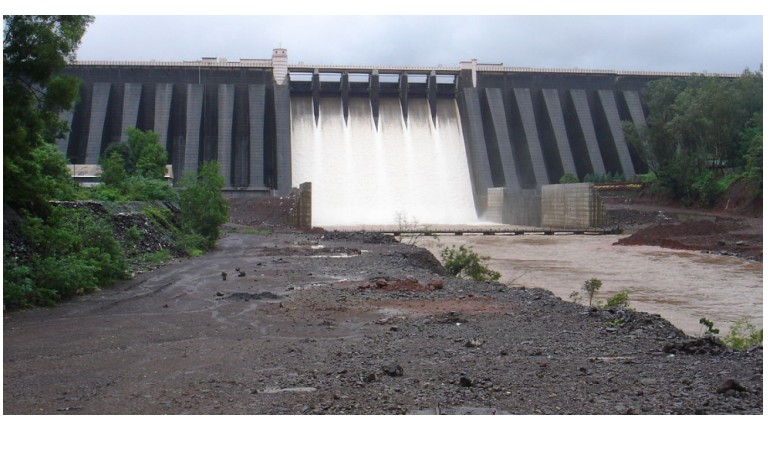

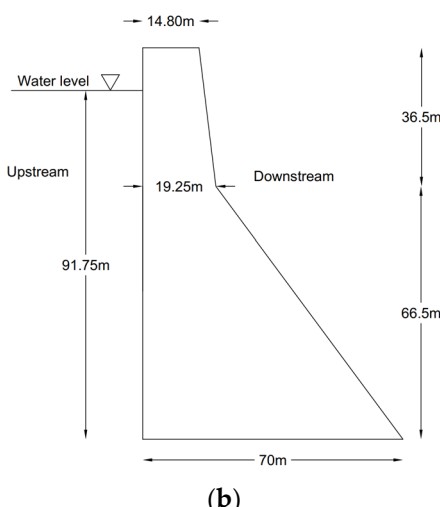

(**a**)

(**b**)

**Figure 6.** Koyna Dam (**a**) downstream view [41], (**b**) cross-section [42].

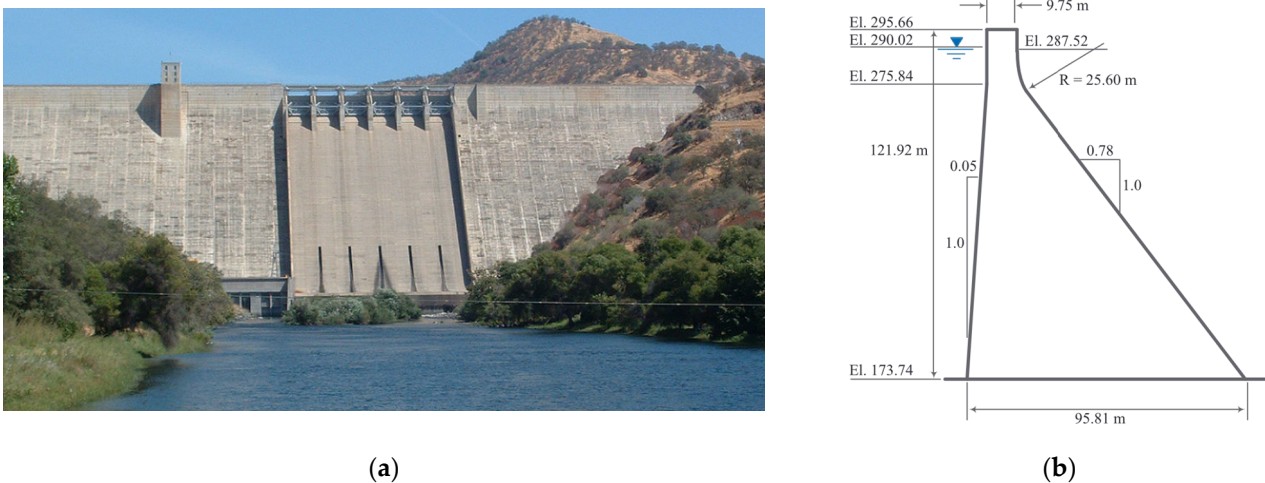

(**a**)  (**b**)

**Figure 7.** Pine Flat Dam (**a**) downstream view [43], (**b**) cross-section [44].

### *4.2. Static and Dynamic Loading*

The analysis accounts for the combined influence of gravity, hydrostatic, and hydrodynamic loading on the system. The seismic response of the Koyna dam was examined using the ground motion data recorded from an accelerograph in one of the galleries of the dam, from the earthquake that occurred on 11 December 1967, with a moment magnitude Mw = 6.5 [22]. The analysis considered ground motion characterized by peak horizontal accelerations of 0.473 g (perpendicular to the dam axis) and peak vertical accelerations of 0.311 g, as illustrated in Figure 8a,b.

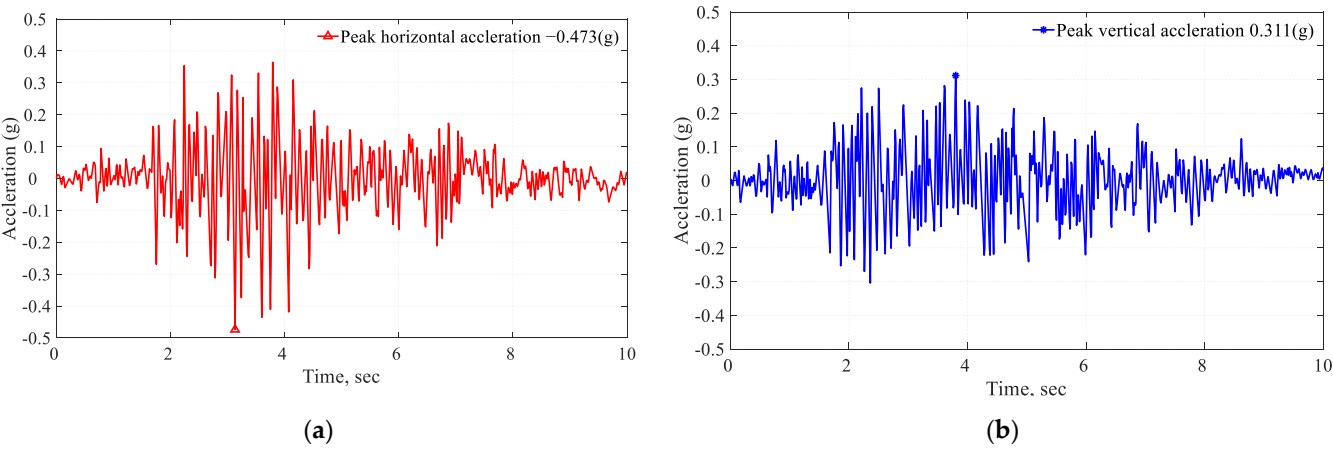

(**a**)  (**b**)

**Figure 8.** Koyna ground motion (**a**) horizontal acceleration and (**b**) vertical acceleration components.

Likewise, in the case of the Pine Flat dam, the seismic analysis focused on the Taft Lincoln School Tunnel earthquake (hereafter referred to as Taft), which occurred on 21 July 1952, with a moment magnitude of Mw = 7.3. Similar to the approach taken for the Koyna dam, the dynamic behavior of the Pine Flat dam's cross-section was evaluated. For this analysis, the ground motion was characterized by peak horizontal accelerations of 0.177 g and peak vertical accelerations of 0.108 g, as depicted in Figure 9a,b [4].

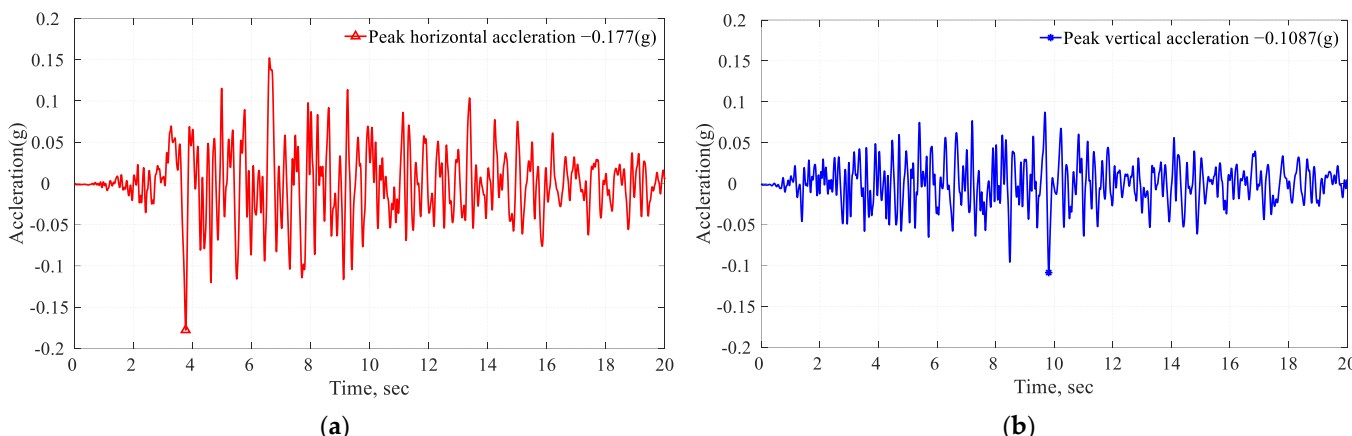

**Figure 9.** Taft ground motion (**a**) horizontal acceleration and (**b**) vertical acceleration components.

### 4.3. Material Properties

Understanding the material properties of the dam, foundation, and reservoir is vital for a comprehensive seismic assessment to ensure an accurate prediction of its behavior and to facilitate informed decision-making for the safety and integrity of dam structures. These properties encompass both general characteristics and specific values utilized in the analysis. While some of the values are typical, others hold specific characteristics pertinent to each dam's behavior. Comparing the material properties of the two dams reveals significant differences. For instance, the Koyna dam exhibits higher modulus of elasticity (E) values than the Pine Flat dam, suggesting a stiffer response. Such disparities can impact structural behavior, influencing factors like vibration damping and deformation. Foundation properties, which reflect the structural interaction with the underlying terrain, affect the energy dissipation during seismic events, affecting how the dam responds to ground motions.

Likewise, reservoir properties, such as the density and wave reflection coefficient, influence the hydrodynamic effects on dam behavior. A higher wave reflection coefficient, as observed in both dams, can amplify hydrodynamic forces. The material properties used in the numerical simulation are shown in Tables 2 and 3.

**Table 2.** Elastic material properties of dam, foundation, and reservoir for Koyna Dam [45].

| Material Properties<br>General and Specific to Abaqus | Dam | Foundation | Reservoir |
|---|---|---|---|
| Density ($\rho$) | 2643 kg/m$^3$ | 2643 kg/m$^3$ | 1000 kg/m$^3$ |
| Modulus of elasticity (E) | 31,027 MPa | 27,580 MPa | - |
| Bulk modulus (K) | - | - | 2070 MPa |
| Poisson's ratio ($\nu$) | 0.15 | 0.333 | - |
| Rayleigh damping Alpha ($\alpha$) | - | 1.64 | - |
| Rayleigh damping Beta ($\beta$) | 0.00323 | 0.0012 | - |
| **Specific to EAGD-84** | | | |
| Wave reflection coefficient ($\alpha$) | - | - | 0.75 |
| Hysteric damping for dam | 0.07 | 0.04 | - |
| **Specific to ADRFS v1** | | | |
| Wave reflection coefficient | - | - | 0.75 |
| Wave speed | - | - | 1440.00 m/s |
| Damping ratio ($\zeta$) | 0.03 | 0.02 | - |

**Table 3.** Elastic material properties of dam, foundation, and reservoir for Pine Flat dam [46].

| Material Properties<br>General and Specific to Abaqus | Concrete | Foundation | Reservoir |
|---|---|---|---|
| Density ($\rho$) | 2482 kg/m$^3$ | 2640 kg/m$^3$ | 1000 kg/m$^3$ |
| Modulus of elasticity (E) | 22,407 MPa | 22,407 MPa | - |
| Bulk modulus (K) | - | - | 2070 MPa |
| Poisson's ratio | 0.2 | 0.333 | - |
| Rayleigh damping Alpha ($\alpha$) | - | 1.64 | - |
| Rayleigh damping Beta ($\beta$) | 0.004333 | 0.00668 | - |
| **Specific to EAGD-84** | | | |
| Wave reflection coefficient ($\alpha$) | - | - | 0.75 |
| Hysteric damping for dam | 0.1 | 0.1 | - |
| **Specific to ADRFS v1** | | | |
| Wave reflection coefficient | - | - | 0.75 |
| Wave speed | - | - | 1440.00 m/s |
| Damping ratio ($\zeta$) | 0.04 | 0.07 | - |

Concrete damage plasticity (CDP) stands out as a widely adopted material model in finite element analysis, offering a means to simulate concrete behavior across a spectrum of loading conditions. It effectively captures inelastic deformation and the accumulation of damage in concrete structures, providing a comprehensive characterization of both tensile and compressive responses, as depicted in Figure 10. However, CDP models fail to capture the change in the material properties based on the stress and deformation states at various stages of a solution. To address that, a modified CDP model based on a Lagrangian formulation was developed by Grassl et al. [47,48]. The concept of modified concrete damage plasticity (CDPM2), as described in [48] and implemented in Abaqus explicit solver [49], incorporates the effects of solution-dependent parameters affecting the material behavior in dynamic loading, high strain rates, cyclic loading, and multi-axial loading, among others.

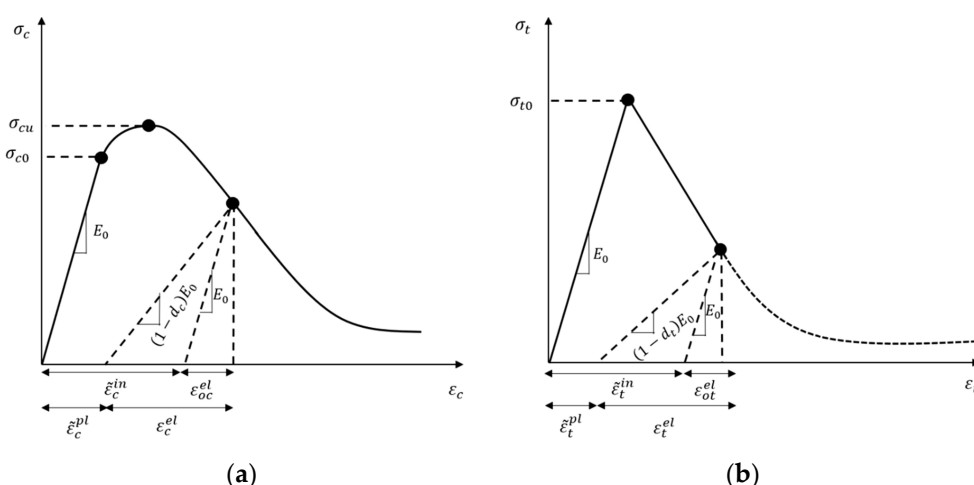

**Figure 10.** Concrete behavior under uniaxial (**a**) compressive and (**b**) tension loading [12].

To understand the effect of material modeling in the solution, six simulations, i.e., two with mean values of the compressive strength and related input parameters for the material models and the remaining four with mean values of the compressive strength $\pm\sigma$ (standard deviation) of these relevant materials parameters, of the dam-only case of Koyna Dam (scenario 4.1.3) were carried out using CDP and CDPM2 models. The explicit solver was used, given that the latter is available only for the explicit solver in Abaqus. The mean

values of the material properties are outlined in Tables 2 and 4. A standard deviation of 35% for $\sigma_{cu}$ is considered based on the work of Rahman Raju et al. [50], which is consistent with the resistance factor for concrete used in design in different codes and standards. Other relevant input parameters such as E, $\sigma_{co}$, and $\sigma_{to}$ are estimated. A comparison of crest displacement histories for both CDP and CDPM2 is shown in Figure 11, where it can be seen that the original CDP model slightly overpredicts the crest displacement as compared to the CDPM2 model, the observed difference is expected as the CDPM2 model is deemed more adaptive. The traditional CDP model produces a slightly conservative response, i.e., 5% to 6% more compared to the CDPM2 model. So, the original CDP model was adopted for all cases of nonlinear analysis and implemented using the implicit solver in Abaqus.

**Table 4.** CDP properties for Koyna and Pine Flat dams.

|  | ψc * | $\sigma_{co}$ (MPa) | $\sigma_{cu}$ (MPa) | $\sigma_{to}$ (MPa) | e | R |
|---|---|---|---|---|---|---|
| Koyna | 36.31° | 13.0 | 24.1 | 2.90 | 0.1 | 1.16 |
| Pine Flat | 36.31° | 12.08 | 22.41 | 2.24 | 0.1 | 1.16 |

* $\psi_c$: dilatation angle; $\sigma_{co}$: compressive initial yield stress; $\sigma_{cu}$: compressive ultimate yield stress; $\sigma_{to}$: tensile failure stress; e: flow potential eccentricity; R: ratio of the initial equibiaxial to the uniaxial compressive yield stress.

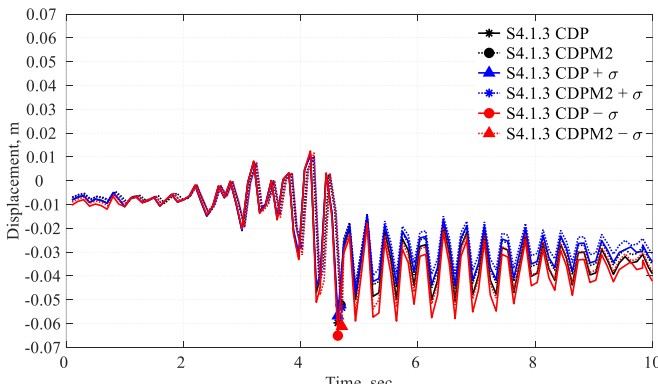

**Figure 11.** Comparison of Koyna Dam horizontal time displacement with CDP and CDPM2.

## 5. Results and Discussion

The subsequent section delves into the discrepancies in modal parameters and crest displacement histories. While the Koyna and Pine Flat dams exhibit geometric similarities, their material properties and loading conditions starkly differ. The present analysis aims to juxtapose these dams, showcasing how these variations impact their responses. We specifically emphasize differences in magnitude and trends within the defined scope of our study, employing diverse solution approaches and model complexities. The intention is not a direct structural comparison, but rather to present these as distinctive case studies of analogous yet individually unique dams.

### 5.1. Modal Analysis

This section explores the modal analysis of the Koyna and Pine Flat dams, shedding light on their dynamic behavior under varying model complexities and solution procedures. Figure 12 presents a comparative analysis of modal periods, offering valuable insights into their distinct dynamic responses. The impact of model complexity, ranging from simple D models to more intricate DFR models on modal periods is examined, as well as how solution procedures influence these periods. This study also examines the variation in modal periods between fundamental and higher modes, elucidating the factors contributing to these distinctions. The analysis focuses on the first four modes of vibration. This choice is based on the fundamental modal period's significance in seismic analysis, representing the primary vibration mode with potential implications for the dams' seismic responses. Moreover, considering higher mode periods is essential to comprehensively understand

the dams' dynamic behavior and assess potential resonance conditions. Key observations regarding the variation in modal periods across different solution procedures and levels of model complexity are highlighted below.

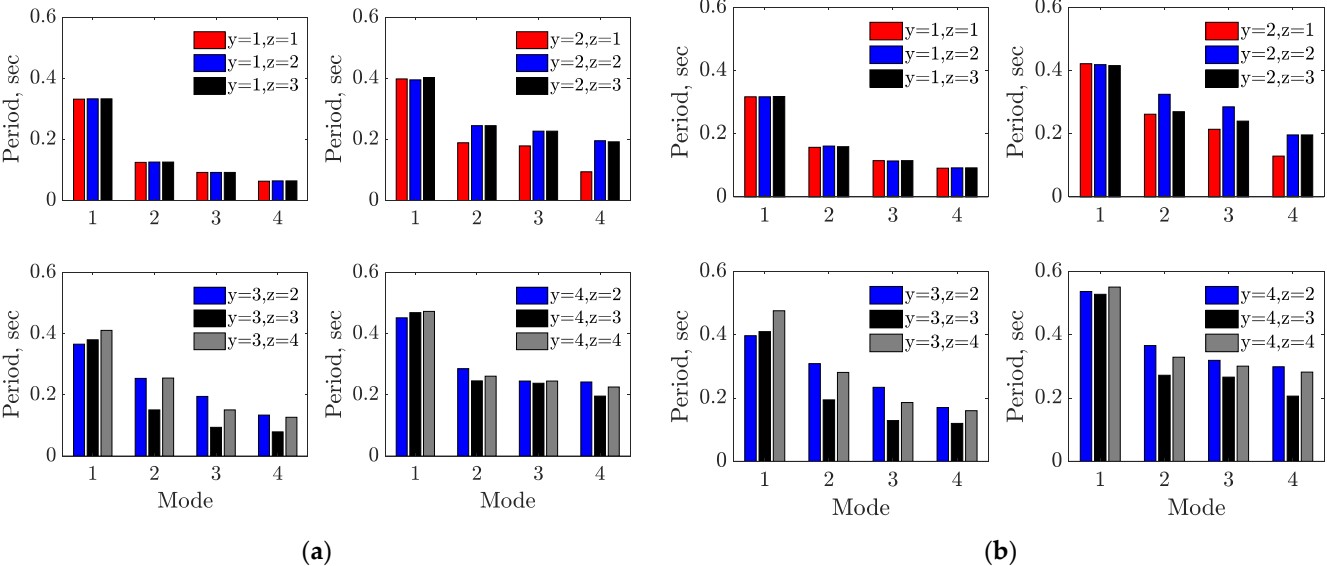

**Figure 12.** Comparison of modal parameters: (**a**) Koyna and (**b**) Pine Flat dams.

### 5.1.1. Comparison of the Modal Period

The fundamental periods for both dams are in a similar range, indicating that their primary vibration modes are comparable. Further, the fundamental period of vibration is relatively shorter for the Koyna dam compared to the Pine Flat dam, indicating that Koyna tends to vibrate at a higher frequency. There is significant variation in higher-mode periods between the two dams, which can be attributed to differences in their structural characteristics, such as size, geometry, and material properties.

### 5.1.2. Impact of Model Complexity

The variation in modal periods with model complexity is due to the added complexities introduced by considering the reservoir and foundation effects. These effects can alter the natural vibration characteristics of the dam. As the model complexity increases from D to DFR, the fundamental period tends to increase for both dams, which is expected as the system incorporates additional elements and complexities. More complex models account for additional structural details and interaction effects, leading to slower vibrations. The variation in modal periods is more pronounced in higher modes, indicating that the effect of model complexity is more significant in these modes.

### 5.1.3. Impact of the Solution Procedure

The choice of solution procedure has a notable influence on the modal periods, with variations observed between the different procedures. In the D models, as expected, the modal periods remain relatively consistent for both dams, regardless of the solution procedure. Across both dams, Abaqus A tends to produce longer fundamental periods compared to other procedures, indicating a less stiff system response. ADRFS v1 also shows variations in the modal periods. However, they are generally shorter than those produced by Abaqus A. EAGD-84 and Abaqus M tend to produce similar modal periods, with some variations depending on the dam and model complexity. Solution procedures with different damping characteristics affect the modal periods, with Abaqus Acoustic introducing higher damping and longer periods.

### 5.1.4. Variation in the Higher Modes

The fundamental period in the DF, DR, and DFR models remains relatively consistent across different solution procedures, indicating that the primary impact on the fundamental period is only due to the increase in model complexity. Higher modes exhibit greater variation in modal periods compared to the fundamental mode. This is because higher modes are associated with more complex and localized deformations within the structure. Small changes in model parameters or solution methods can have a larger impact on these modes. Further, the variation in modal periods can be attributed to the inherent differences in how each software package formulates and solves the dynamic equations of motion. Additionally, the choice of elements, integration schemes, and convergence criteria can influence the results.

### 5.2. Crest Displacement

When evaluating variations in the system response, it is essential to consider several key aspects. Firstly, one should assess model complexity, encompassing D, DF, DR, and DFR, to comprehend its influence on crest displacement. Secondly, the impact of different solution procedures should be compared to detect any disparities or inconsistencies in the results. Thirdly, an examination of various reservoir modeling techniques, seismic load application scenarios, a comparison of linear and nonlinear analyses, and an assessment of computational efficiency are crucial. By considering these factors, a comprehensive evaluation was conducted to gain insights into the variations in the system response and the underlying factors contributing to them.

### 5.2.1. Mean and Standard Deviation for S1 and S2

The mean ($\mu$) and standard deviation ($\sigma$) values for the maximum crest displacement as obtained from the three software systems corresponding to each model complexity for S1 and S2 are depicted in Figure 13. These values are examined across various model complexities and seismic scenarios, offering the following insights into the seismic responses of the Koyna and Pine Flat dams:

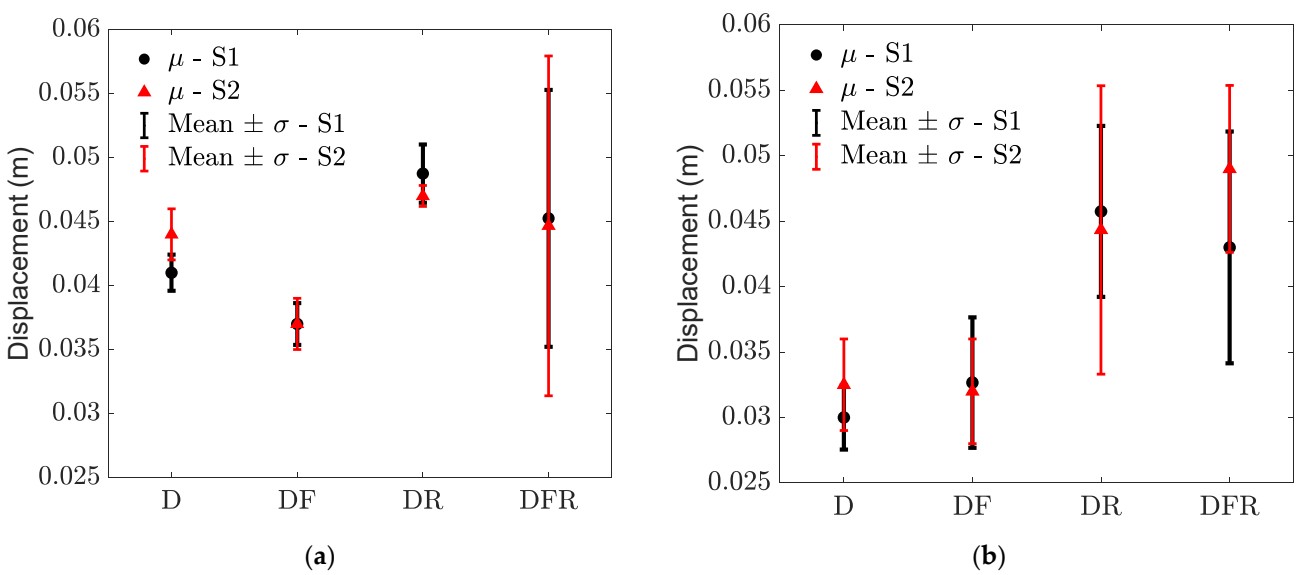

**Figure 13.** Displacement variation among software tools for different model complexities: (**a**) Koyna Dam and (**b**) Pine Flat Dam.

As the models become more complex, incorporating factors like DR and DFR interactions, the mean displacement values tend to increase. This phenomenon is observed in both dams, indicating that the inclusion of additional complexities leads to larger average displacements. Also, the standard deviation tends to increase with higher model complexity.

This suggests that more complex models introduce greater variability in the dam's seismic response, potentially due to the intricate features considered.

When comparing the two dams, Koyna Dam consistently exhibits higher mean displacement values than Pine Flat. This discrepancy arises from the inherent structural and material differences between the dams. However, the standard deviation for Koyna Dam is generally smaller than that of Pine Flat Dam, indicating that Koyna's responses are more consistent and less variable, except in the most complex scenario.

The variation across model complexities is most pronounced in the most complex model, DFR, which considers a wide range of intricate factors. This model introduces the greatest variability in the results, highlighting the importance of carefully selecting the appropriate level of complexity for seismic analysis.

5.2.2. Displacement Histories for S1 and S2

Table 5 presents the maximum crest displacement across different solution procedures and model complexities. Figures 14 and 15 showcase the displacement time histories for S1, illustrating the variations in trends and magnitudes. The crest displacement trends and values for both dams closely match across the three software tools in the D model, as shown in Figures 14a and 15a. For the DF model in Koyna, the trend exhibits a close match with minor variations in magnitude, as depicted in Figure 14b. However, the time of occurrence of the maximum displacement differs among the three solution procedures. These discrepancies become more pronounced in the DF model for Pine Flat, as observed in Figure 15b. In the case of the DR and DFR models for Koyna, the Abaqus A and ADRFS v1 solutions yield the most consistent and least variable results, while the Abaqus M and EAGD-84 solutions exhibit higher variation, as seen in Figure 14c,d. Correspondingly, for Pine Flat, Abaqus A, ADRFS v1, and Abaqus M solutions consistently provide consistent and less variable results, while the EAGD-84 solution exhibits higher variation, as shown in Figure 15c,d. The variation in results between the two dams suggests that the selection of solution procedure remains critical. Further, the specific behavior of each dam under seismic conditions is influenced by factors such as geometry, material properties, and loading conditions.

**Table 5.** Maximum crest displacement in (m) for scenarios S1 and S2 across different solution procedures and model complexities.

| Scenario (x =) | Model Complexity (y =) | Koyna Dam | | | | Pine Flat Dam | | | |
|---|---|---|---|---|---|---|---|---|---|
| | | Software (z =) | | | | | | | |
| | | 1 | 2 | 3 | 4 | 1 | 2 | 3 | 4 |
| S1 | 1 | 0.042 | 0.042 | 0.039 | - | 0.027 | 0.030 | 0.033 | - |
| S1 | 2 | 0.037 | 0.039 | 0.035 | - | 0.026 | 0.038 | 0.034 | - |
| S1 | 3 | 0.045 | 0.051 | 0.049 | 0.050 | 0.041 | 0.043 | 0.057 | 0.042 |
| S1 | 4 | 0.056 | 0.046 | 0.029 | 0.050 | 0.031 | 0.043 | 0.056 | 0.042 |
| S2 | 1 | 0.046 | - | 0.042 | - | 0.029 | - | 0.036 | - |
| S2 | 2 | 0.035 | - | 0.039 | - | 0.028 | - | 0.036 | - |
| S2 | 3 | 0.047 | - | 0.046 | 0.048 | 0.031 | - | 0.058 | 0.044 |
| S2 | 4 | 0.059 | - | 0.027 | 0.048 | 0.045 | - | 0.058 | 0.044 |

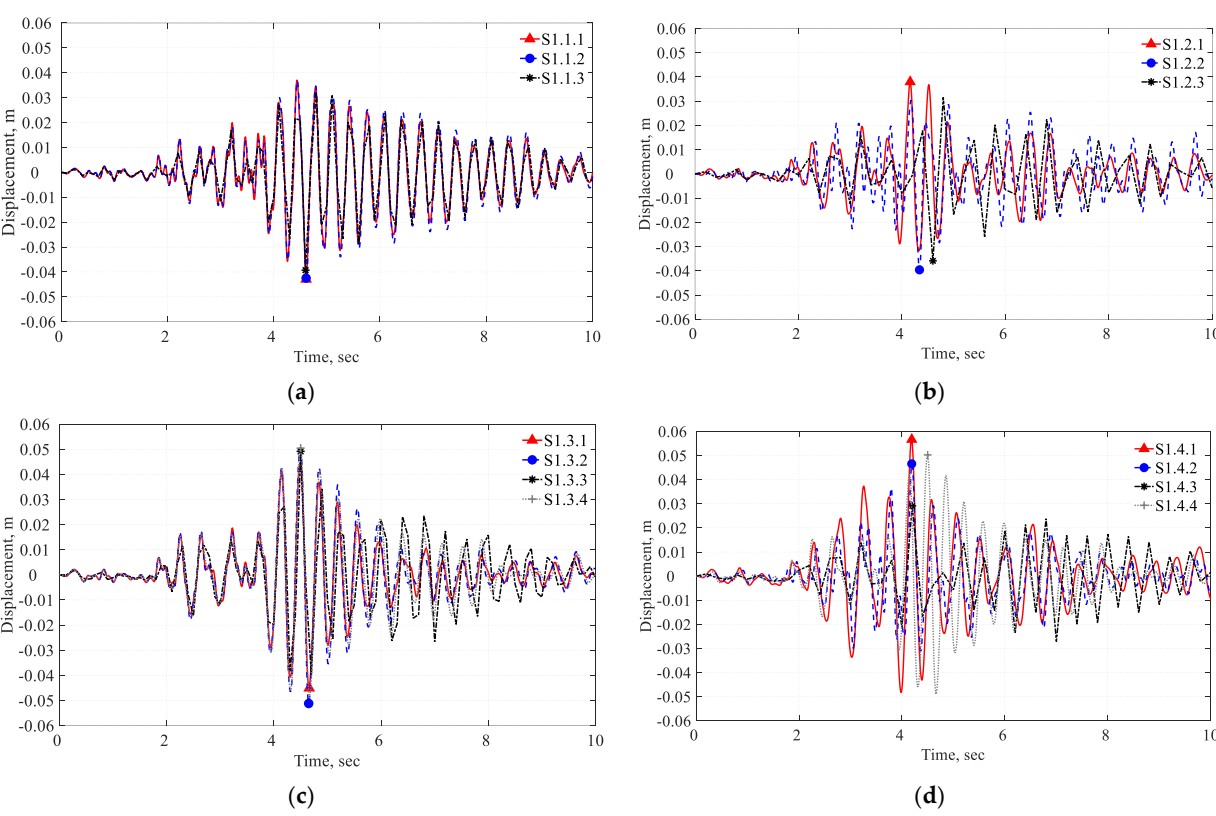

**Figure 14.** Koyna Dam horizontal time displacement for S1 scenario: (**a**) D, (**b**) DF, (**c**) DR, and (**d**) DFR.

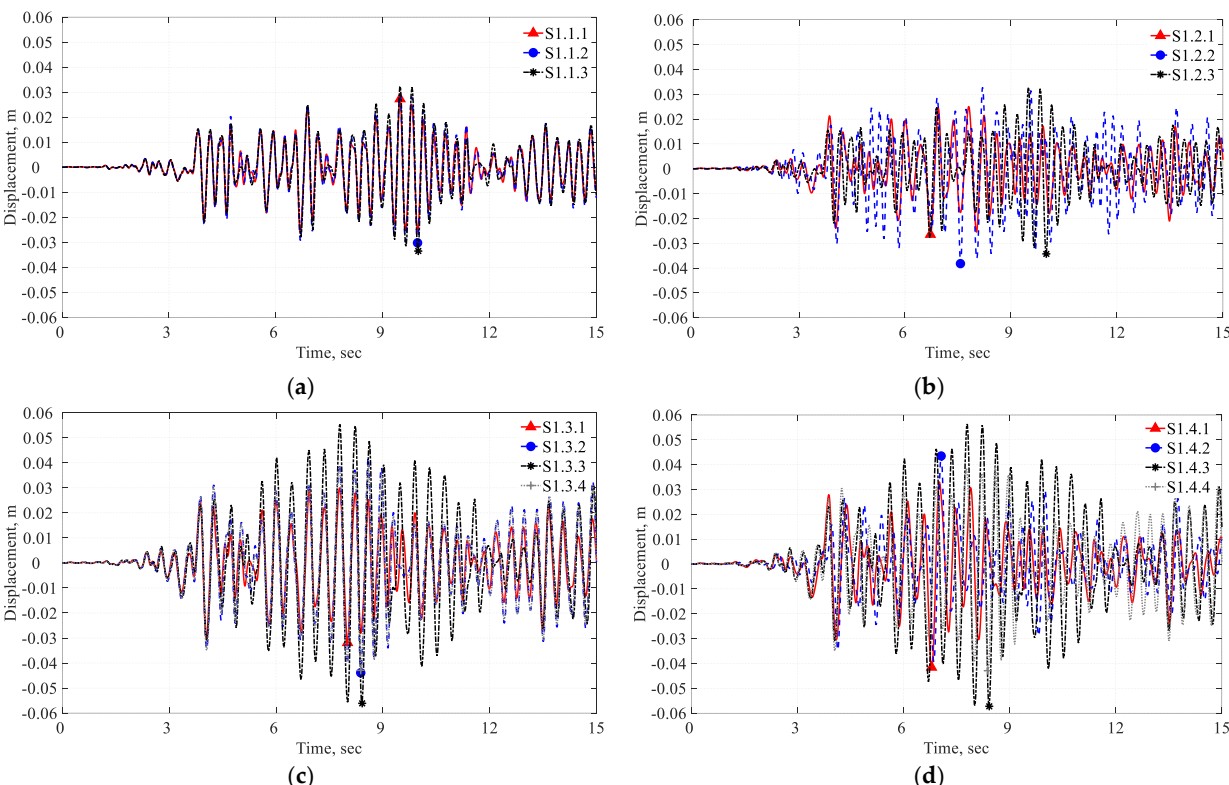

**Figure 15.** Pine Flat horizontal time displacement for S1 scenario: (**a**) D, (**b**) DF, (**c**) DR, and (**d**) DFR.

In the same manner, Figures 16 and 17 depict the displacement histories for scenario S2. Similar to S1, when only the body of the dam is considered in the model, the crest displacement trends and magnitudes closely match across all the software tools for both dams, as shown in Figures 16a and 17a. However, for Pine Flat Dam, the time of occurrence of maximum displacement is slightly different across the software tools, as can be observed in Figure 17a. For the Koyna DF model, there is a close match in trend and magnitude, although the maximum displacement occurs at different periods, as depicted in Figure 16b. Likewise, for Pine Flat Dam, there is a trend-wise match, but the variations in magnitude and the time when the maximum displacement occurs are more significant, as shown in Figure 17b. In the DR model, a consistency in magnitude and trend can be observed for the Koyna dam in Figure 16c, while significant discrepancies are evident for the Pine Flat dam in Figure 17c. The DFR model exhibits consistent trends but substantial variations in magnitude for both dams, as depicted in Figures 16d and 17d. The Abaqus A solution provides the most consistent response across the solution procedures for both dams, regardless of the model complexity.

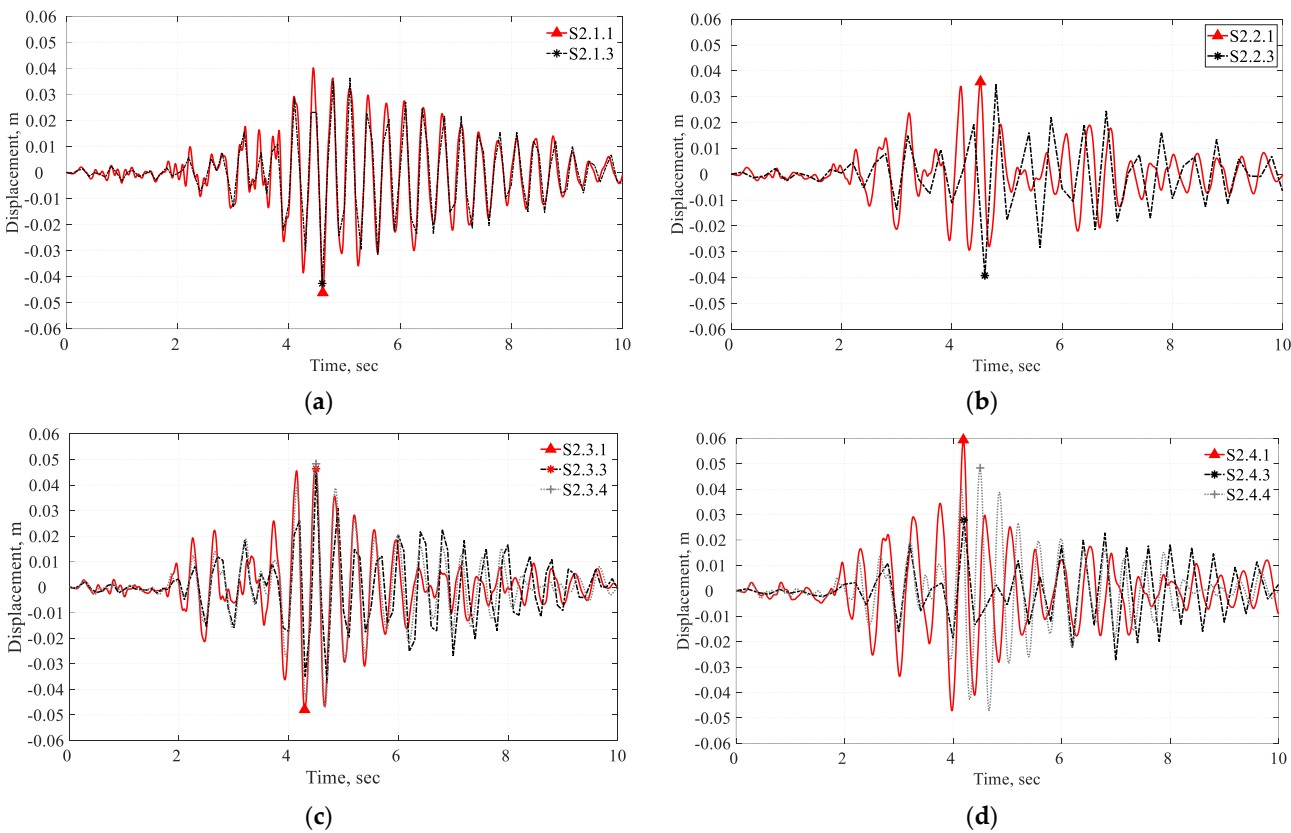

**Figure 16.** Koyna Dam horizontal time displacement for S2 scenario: (**a**) D, (**b**) DF, (**c**) DR, and (**d**) DFR.

### 5.2.3. Displacement Histories for S3 and S4

In order to further evaluate the variation between the different modeling approaches, a comparison between linear and nonlinear analysis was also conducted as shown in Table 6. When comparing scenarios S3 and S4, Table 6 demonstrates a notable increase in displacement magnitude within the D and DF models for both dams. This discrepancy underscores the significance of accounting for both horizontal and vertical ground motion components. The elevated crest displacement observed in scenario S4, in contrast to scenario S3, can be attributed to the influence of the vertical seismic component. When a dam experiences both horizontal and vertical seismic forces, their complex interaction can produce an anti-gravity-like effect, temporarily reducing the dam's effective weight. Consequently, the dam exhibits higher displacements due to a heightened dynamic response.

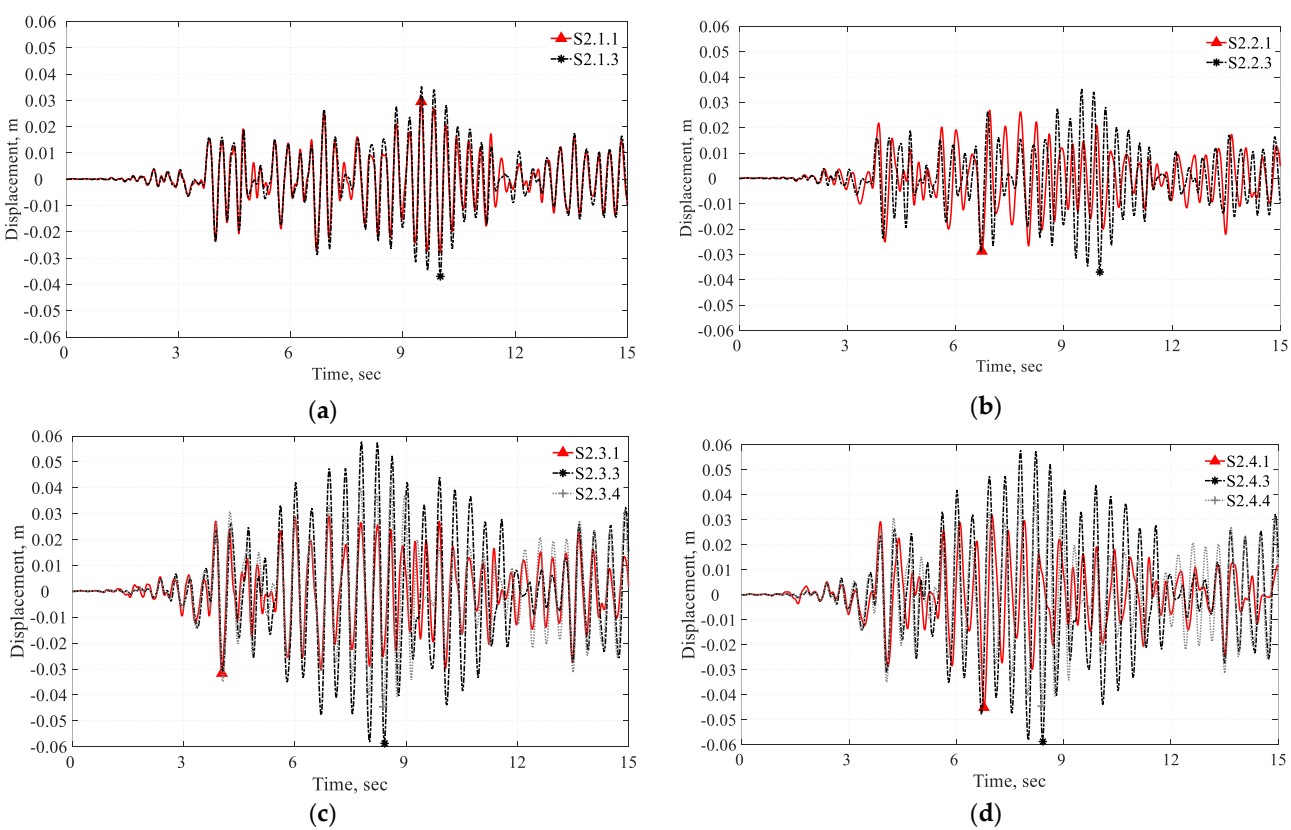

**Figure 17.** Pine Flat Dam horizontal time displacement for S2 scenario: (**a**) D, (**b**) DF, (**c**) DR, and (**d**) DFR.

**Table 6.** Maximum crest displacement in (m) for scenarios S3 and S4 across different solution procedures and model complexities.

| Scenario (x =) | Model Complexity (y =) | Koyna | | Pine Flat | |
|---|---|---|---|---|---|
| | | Software (z =) | | | |
| | | **3** | **4** | **3** | **4** |
| S3 | 1 | 0.040 | - | 0.035 | - |
| S3 | 2 | 0.042 | - | 0.035 | - |
| S3 | 3 | 0.035 | 0.043 | 0.066 | 0.043 |
| S3 | 4 | 0.037 | 0.044 | 0.077 | 0.044 |
| S4 | 1 | 0.051 | - | 0.043 | - |
| S4 | 2 | 0.047 | - | 0.043 | - |
| S4 | 3 | 0.035 | 0.040 | 0.054 | 0.044 |
| S4 | 4 | 0.037 | 0.040 | 0.077 | 0.046 |

As in the previous section, Figures 18 and 19 present the displacement time histories for scenario S3. For Koyna Dam, in the D and the DF model, more inelastic behavior can be observed in the nonlinear analysis compared to the linear analysis given the difference between both curves as depicted in Figure 18a,b. Conversely, for Pine Flat Dam, in the D and DF models, the trends and magnitudes match closely between linear and nonlinear analysis, as shown in Figure 19a,b, indicating that the dynamic behavior of the dam during the Taft earthquake remains in the elastic range. In the DR and DFR models for the Koyna dam, the magnitude of displacement is lower in the nonlinear analysis compared to the linear analysis, as illustrated in Figure 18c,d. On the contrary, for Pine Flat Dam, the magnitude of displacement is higher in the nonlinear analysis compared to the linear analysis, as depicted in Figure 19c,d. It is worth noting that the observed displacement values in the

nonlinear analysis can be either higher or lower compared to the linear analysis. The lower displacement in nonlinear analysis can be attributed to possible damage and stress redistribution in the structure. Moreover, it is observed that in the case of Koyna dam, the nonlinear displacement values are generally lower than the linear analysis, indicating a closer conformity to the reported damage state in the real structure. However, for Pine Flat Dam, both linear and nonlinear analyses show similar displacement values, indicating elastic behavior and less damage. Variations in displacement can also be observed in the DR and DFR models, particularly with the Abaqus M in both linear and nonlinear analyses. However, the Abaqus A solution provides consistent results across both linear and nonlinear analyses.

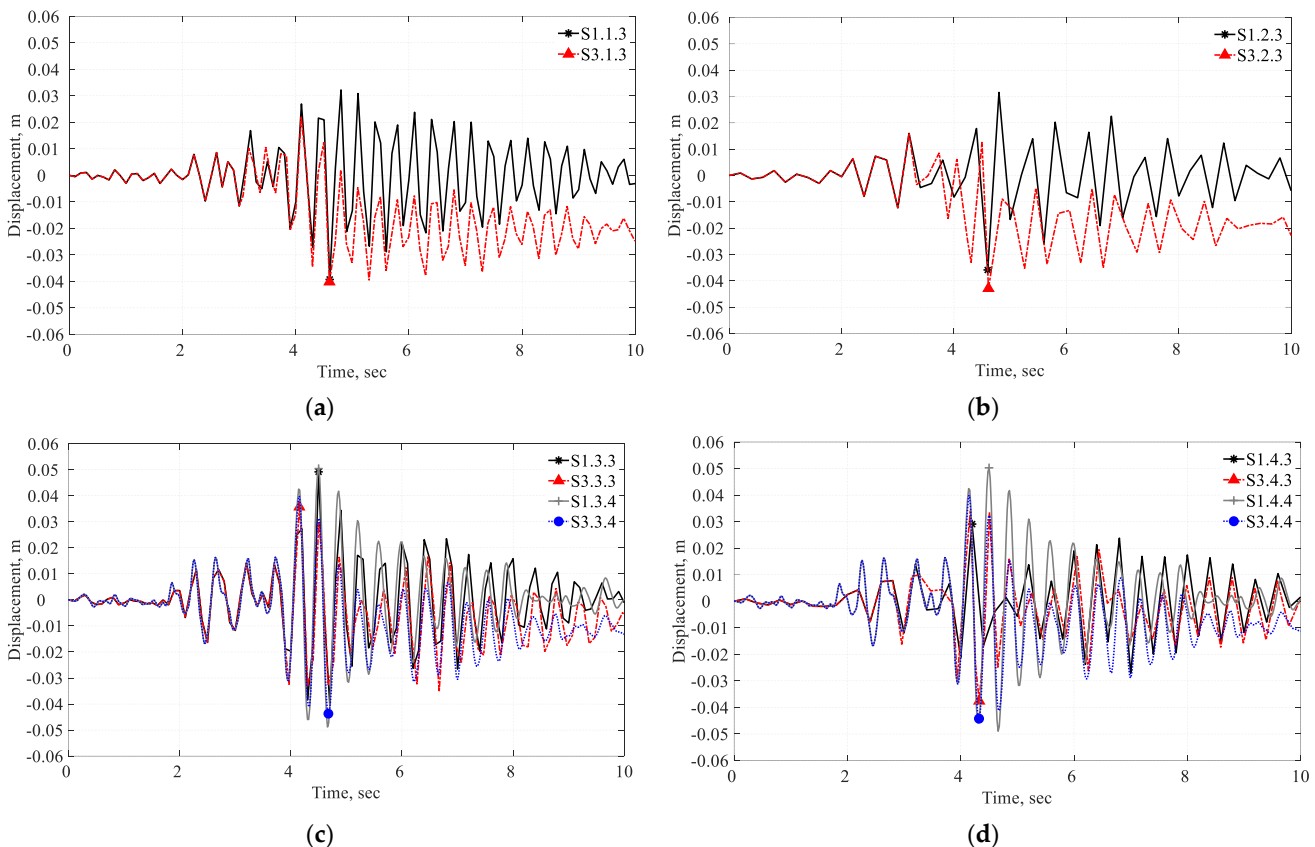

**Figure 18.** Koyna Dam horizontal time displacement: (**a**) D, (**b**) DF, (**c**) DR, and (**d**) DFR, linear (S1) vs. nonlinear analysis (S3) considering horizontal ground motion component only.

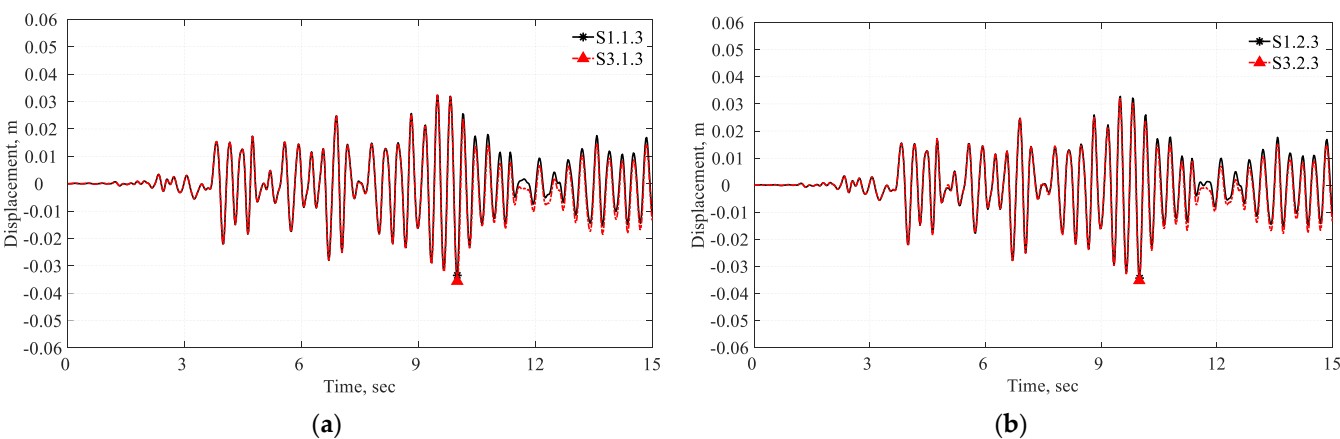

**Figure 19.** *Cont.*

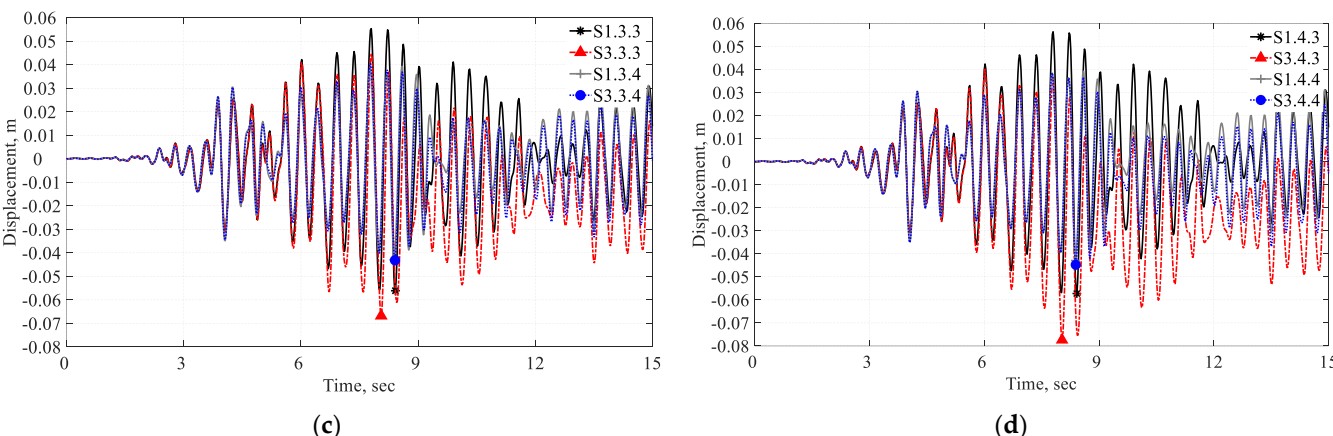

(**c**)  (**d**)

**Figure 19.** Pine Flat Dam horizontal time displacement: (**a**) D, (**b**) DF, (**c**) DR, and (**d**) DFR, linear (S1) vs. nonlinear analysis (S3) considering the horizontal ground motion component.

It is also equally important to note that the observed inelastic behavior of the D and DF models as illustrated in Figure 20a,b. Figure 21a,b show the criticality of the reservoir-empty condition. Further, with the reservoir-filled condition, there is an elevated displacement in the DFR model of Pine Flat Dam as compared to the Koyna dam. So, the dynamic response is intensified in Pine Flat Dam; however, in the case of Koyna, it is damped.

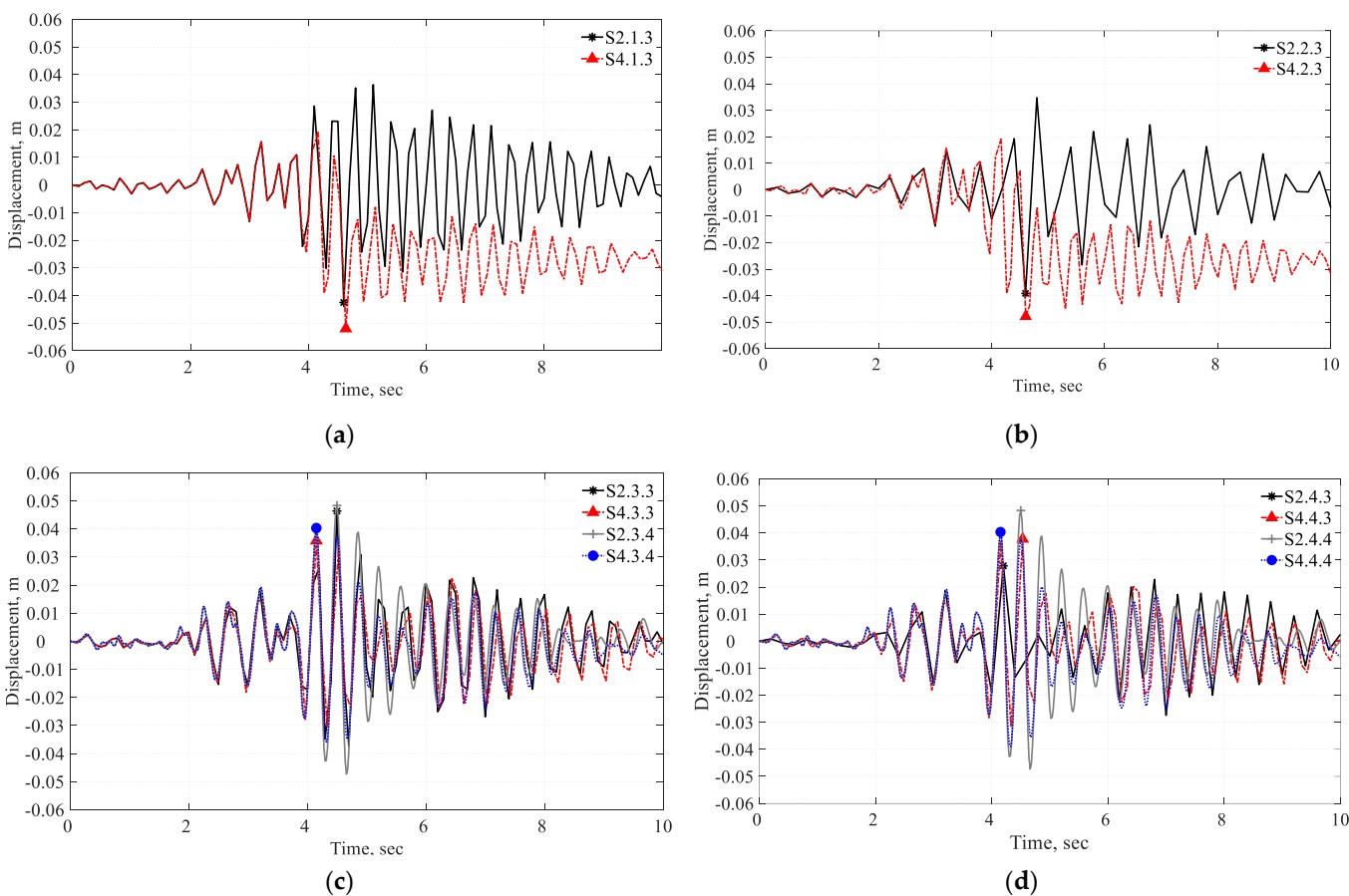

(**a**)  (**b**)

(**c**)  (**d**)

**Figure 20.** Koyna Dam horizontal time displacement: (**a**) D, (**b**) DF, (**c**) DR, and (**d**) DFR, linear (S2) vs. nonlinear analysis (S4) considering horizontal and vertical ground motion components.

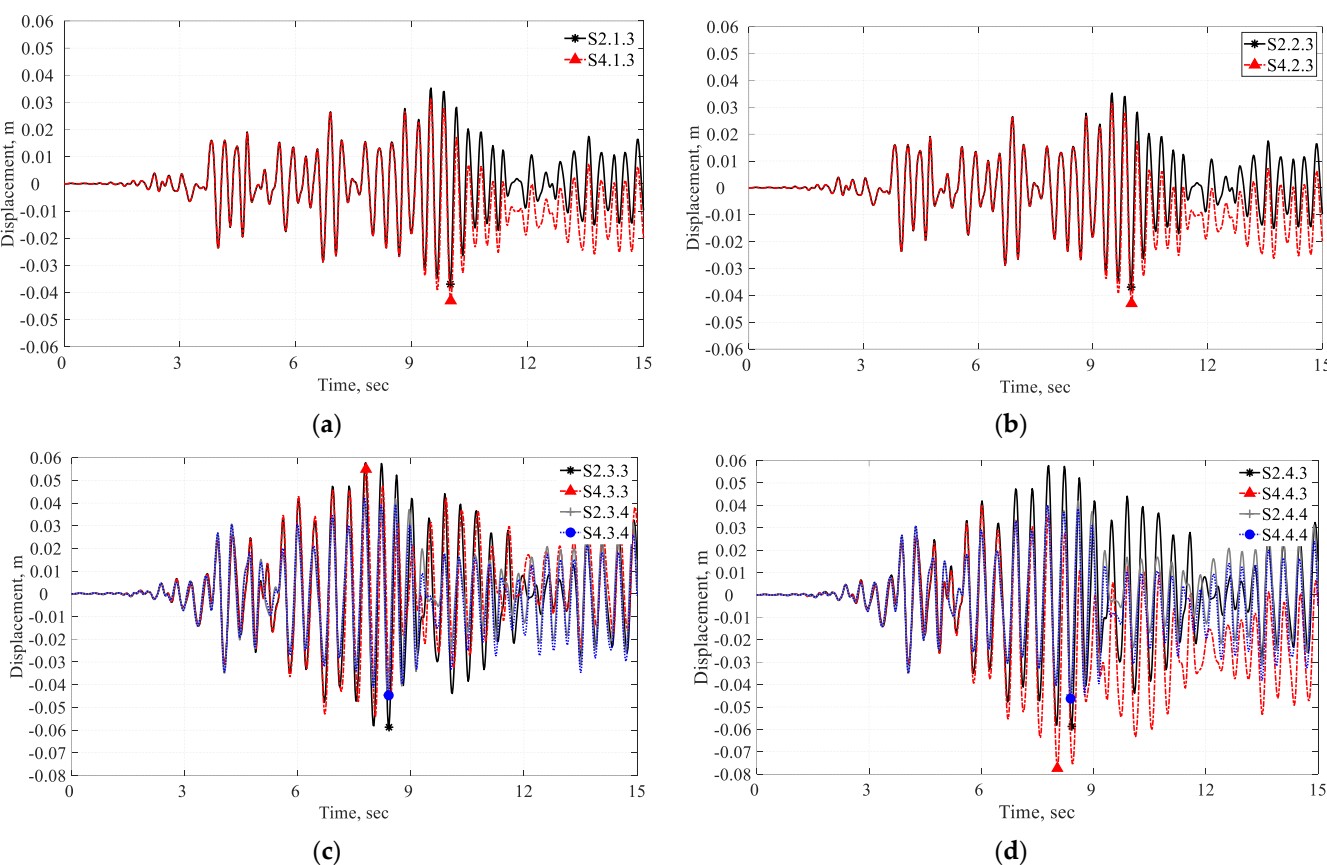

**Figure 21.** Pine Flat Dam horizontal time displacement: (**a**) D, (**b**) DF, (**c**) DR, and (**d**) DFR, linear (S2) vs. nonlinear analysis (S4) considering horizontal and vertical ground motion components.

Figures 22 and 23 summarize the key findings of the section, comparing the maximum crest displacement across scenarios S1 to S4. From Figure 22, it can be observed that while there are discrepancies between all four scenarios, variation in the results is more evident for Pine Flat than for Koyna. Additionally, it is shown that the influence of vertical ground motion appears to have a relatively minor impact on displacement values when compared between scenarios, taking into account the nonlinearities and increasing model complexity.

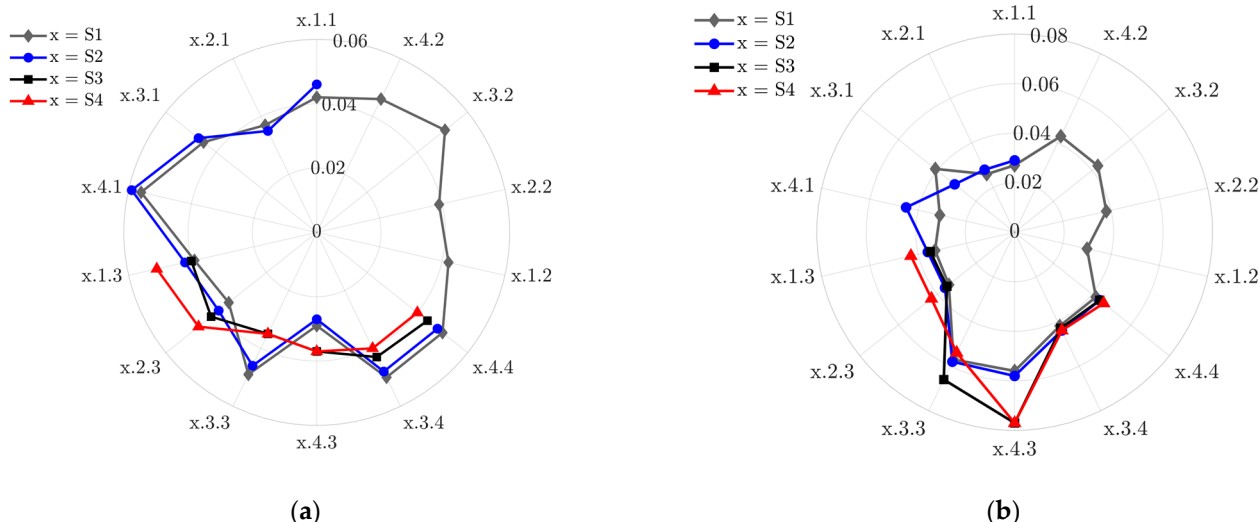

**Figure 22.** Crest displacement (**a**) Koyna Dam, (**b**) Pine Flat Dam.

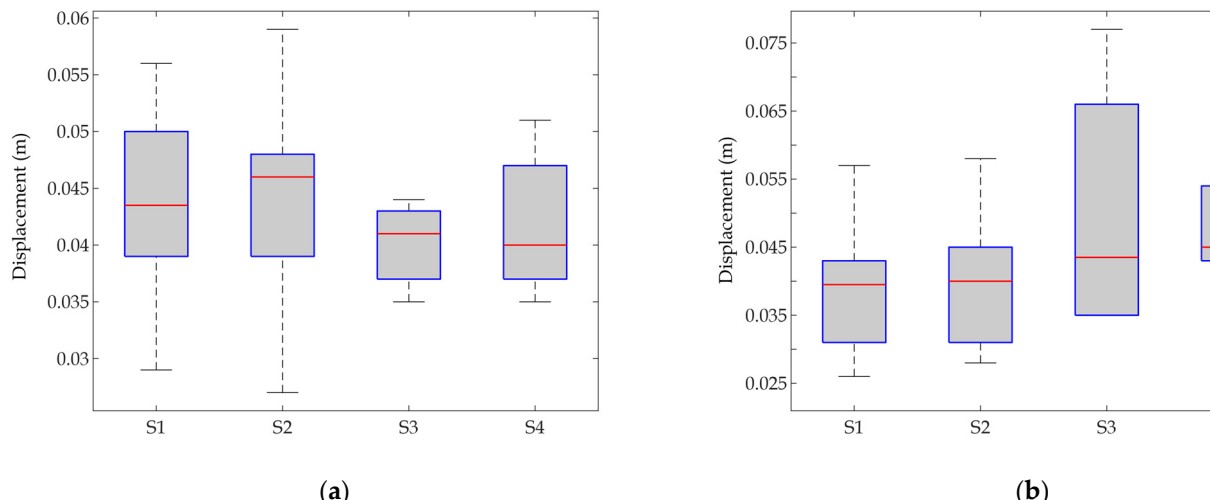

**Figure 23.** Displacement variation among scenarios, (**a**) Koyna Dam, (**b**) Pine Flat Dam.

From Figure 23, the displacement variation for both the Koyna and Pine Flat dams across all considered scenarios can be observed. For both dams, the presence of nonlinearity significantly impacts the crest displacement, with Pine Flat Dam showing a wider range and higher variability. These findings underscore the importance of supplementing linear analysis with nonlinear approaches to attain a realistic understanding of structural responses, especially when dealing with seismic conditions.

### 5.3. Comparison among Software Tools

In addition to the variations in seismic analysis results discussed earlier, it is essential to consider the computational burden imposed by each software, as this factor can significantly influence the feasibility of using a particular software for routine safety assessments versus critical, in-depth analyses. When it comes to ease of use, EAGD-84 is the most accessible, followed by ADRFS v1, while Abaqus is considered more complex. However, the key factor to note is the time required for simulation. EAGD-84 typically offers the shortest analysis time ranging from 20 to 30 s per simulation, making it suitable for routine safety assessments where quick evaluations are essential. ADRFS v1 falls in the middle in terms of computational time, ranging from 40 to 60 s per simulation.

In contrast, Abaqus, while powerful and precise, often demands more time, i.e., ranging from 90 to 150 s per analysis. Therefore, it might be reserved for critical cases or detailed assessments where computational resources are a manageable factor. Engineers should carefully weigh the software's computational demands against their specific analysis requirements, available resources, and the urgency of the assessment to select the most suitable tool for ensuring both safety and efficiency in dam safety evaluations.

### 5.4. Verification with ICOLD Benchmark Study

A recent study presented in [7] addresses the modeling variabilities in seismic assessment of dams quite exhaustively. In that study, results from a set of twenty DFR models (2D, 3D slice, and full 3D) of a benchmark problem from the 15th ICOLD International Benchmark Workshop and the 2018 USSD Benchmark Workshop were used in evaluating the modeling variability and uncertainty [8,9]. While the present study considers twenty-four different models, with increasing complexity, it is synergistic with the study presented in [7] and the results are complementary. For the benchmark concrete gravity dam problem on Pine Flat Dam studied in [8], the results of the modal analysis and the dam crest displacement variations found in the present study are quite consistent for comparable scenarios with those reported in [8], considering the variabilities in the input parameters (e.g., slight disparity in reservoir level, foundation dimensions, and material properties) and modeling assumptions inherent to each software tool used here. It is important to

note the disparities in material properties, specifically $\sigma_{cu}$, and $\sigma_{to}$, between the current article (22.41 MPa and 2.24 MPa, respectively) and those used in [7] (28.0 MPa and 2.0 MPa, respectively). Similarly, variations in foundation geometry are evident, with the current article using dimensions of 412 m by 206 m, contrasting with [7] where dimensions of 700 m by 122 m were used. The reservoir level considered in the current study is 290.02 m vs. 278.57 m in Case A2 [8]. Lastly, regarding differences in Rayleigh viscous damping parameters for the dam, the current article employs $\alpha = 0$ and $\beta = 0.004333$, while [7] uses $\alpha = 0.751$ and $\beta = 0.0005$.

A cross-validation of the findings for comparable cases of the current study was made with [8]. The outcomes for comparable scenarios and the results of the current study fall within the broad range of results presented in [8]. Table 7 shows the comparison of modal parameters for DFR models (scenarios, y = 4, z = 3 and y = 4, z = 4) of the present article with Case A2 presented in [8].

**Table 7.** Comparison of modal parameters with the 15th ICOLD International Benchmark Workshop [8].

| Mode | Natural Frequency (Hz) | | | St. DeviationICOLD Benchmark [3] |
|---|---|---|---|---|
| | Present Study [1] | Present Study [2] | ICOLD Benchmark [3] | |
| 1 | 1.90 | 1.81 | 2.15 | 0.34 |
| 2 | 3.67 | 3.04 | 3.28 | 0.63 |
| 3 | 3.76 | 3.32 | 3.91 | 0.79 |
| 4 | 4.86 | 3.54 | 4.51 | 0.88 |

[1] scenario, y = 4, z = 3 and [2] scenario, y = 4, z = 4 and [3] Case A2 of 15th ICOLD International Benchmark Workshop.

Scenario S1.4.4 in the current article matches with Case D-3 in [8]. The maximum crest displacement for scenario S1.4.4 (0.042 m) of the current article is within the reported range for case D-3 [8]. Similarly, the scenario S3.4.4 in this article matches with the case E-1 in [8]. The maximum crest displacement values for S3.4.4 (0.044 m) align with E-1 and fall within the reported range [8]. The reasonable variation in outcomes can be attributed to the differences in the considered material properties, reservoir levels, and foundation dimensions. Further, there is variation in the modeling approach adopted by participants. A large number of models can provide an improved understanding and quantification of the modeling uncertainty than a case where fewer analysis models are used. However, fewer, but carefully selected models for progressive analysis with increasing complexity could be helpful when there is some a priori knowledge of the modeling uncertainty and the degree of accuracy required.

## 6. Conclusions

Numerical simulation plays a crucial role in assessing the seismic performance of dams, but the choice of solution procedure and model complexity can lead to a wider variation in results. In this context, the current parametric study focused on two selected case study dams with similar geometry but different material and loading characteristics to understand variations in system responses. The selected dams were analyzed for four scenarios (S1 to S4) with increasing model complexity and varying solution procedures. A systematic comparison was conducted to understand the impact on the system response by studying the variations in modal parameters and crest displacement across the different scenarios. The results of the present study are found to be consistent with the findings for the ICOLD benchmark studies, indicating that a detailed individual study with consistent modeling assumptions could produce an outcome within the broad range of the outcome of a set of detailed studies as reported in [7,8].

The observations within the considered premise of the detailed parametric study presented here led to the following key findings:

(i) For both dams, in the case of the dam-only model, the modal parameters and crest displacement histories match quite well across all three software systems in scenarios S1 and S2.

(ii) The observed discrepancy in the crest displacement values between the Westergaard-added mass approach and the reservoir modeled using acoustic elements in the DFR models for the Koyna and Pine Flat dams highlights the impact of using simplified approaches such as the added mass approach. Specifically, in the case of Koyna, the Westergaard-added mass approach yields values of the crest displacement approximately 40% lower than those obtained using the acoustic elements, for both scenarios S1 and S2. Conversely, for Pine Flat, the same approach reports 33% higher values.

(iii) The reservoir, when modeled using acoustic elements, captures fluid–structure interaction more accurately than the Westergaard-added mass approach, which considers only a fraction of reservoir participation. This distinction led to an observed increase in the vibration period when acoustic elements were employed. This indicates that the model with acoustic elements attracts lower seismic force compared to the added mass model. Thus, careful selection of reservoir models for fluid–structure interaction is essential.

(iv) Although isotropic and homogeneous material properties and boundary conditions were adopted for the foundation, a variation in results across the DF models for scenarios S1 and S2 in all three software tools was observed. This variation can be attributed to the way each tool implements the soil–structure interaction.

(v) The maximum crest displacement showed increasing variation with increasing modeling complexity, emphasizing the importance of progressive simulation with increasing model complexity for a proper understanding of the system's behavior.

(vi) In the case of scenarios S3 and S4 for both the D and DF models, with the reservoir empty condition, the effect of nonlinear analysis was more pronounced in the Koyna dam. An empty reservoir condition is often more critical during seismic events as it lacks the damping effects of water, potentially leading to more significant structural stress and displacement in concrete gravity dams.

(vii) Similarly, in the DR and DFR models, the effect of nonlinear analysis was more pronounced for the reservoir-filled condition in the case of Pine Flat Dam; this behavior can be attributed to its dynamic properties.

(viii) The reservoir modeled using acoustic elements in the case of DR and DFR models provided the most consistent results in all four scenarios across both the dams.

(ix) Nonlinear effects were more pronounced for Koyna than for Pine Flat Dam, highlighting that while providing a more adequate response, the use depends on the level of seismic safety evaluation desired.

(x) EAGD-84 and ADRFS v1 are easy to use, and quicker-turnaround tools can be helpful for a preliminary safety assessment. For comprehensive safety assessments, however, Abaqus or similar software remains a suitable choice for a detailed analysis.

(xi) Both EAGD-84 and ADRFS v1 utilize a pressure-based formulation for reservoir modeling, akin to Abaqus acoustic elements. Therefore, for linear assessments, these tools can be a suitable alternative to Abaqus.

In light of the findings from this comprehensive parametric study, several key recommendations emerge to guide dam safety assessments. For initial screenings or routine evaluations of well-maintained dams, simpler methods like 2D modeling using EAGD-84 and ADRFS v1 are expected to be quite effective, offering quick insights into potential safety concerns. Similarly, compliance with regulatory standards such as USBR (2006) [19] necessitates a mix of methods, with initial assessments employing simpler models but escalating to more complex analyses if issues are identified. This is especially true for dams with high-risk profiles, emphasizing the need for a nuanced understanding of variations in system responses. In the design and retrofitting stages, embracing detailed analyses with Abaqus or similar tools is essential to ensure structural resilience against potential extreme events.

Understanding variations across different methods is paramount. The discrepancies found in this study, particularly in crest displacement values between simplified approaches like the added mass method and the more accurate acoustic element modeling, highlight the potential pitfalls of relying on overly simplistic methods. These variations can significantly impact safety assessments, potentially leading to either overly conservative or underestimated response parameters. Therefore, a nuanced approach, as demonstrated by the consistent outcomes provided by the DR or DFR model with acoustic elements across different dams, minimizes variations and instills confidence in practitioners. This consistency is particularly crucial in emergency response planning, where simpler models can rapidly assess potential impacts, contributing to the development of effective action plans. In essence, the choice of linear or nonlinear models and 2D or 3D representations should be dictated by the dam's condition, potential risks, and the desired depth of analysis, with a constant awareness of the implications of these choices on safety evaluations. Finally, future studies can further consider factors such as joint opening/closing at lift joints, base sliding, nonlinear contact modeling, and heterogeneous foundation simulation to comprehend the variations in the system response and suggest consistent modeling techniques.

**Author Contributions:** Conceptualization, B.K.P. and A.B.; methodology, B.K.P.; software, B.K.P.; validation, B.K.P., R.L.S. and A.B.; formal analysis, B.K.P.; investigation, B.K.P.; resources, A.B.; data curation, B.K.P.; writing—original draft preparation, B.K.P.; writing—review and editing, R.L.S. and A.B.; visualization, B.K.P. and R.L.S.; supervision, R.L.S. and A.B.; project administration, A.B.; funding acquisition, A.B. All authors have read and agreed to the published version of the manuscript.

**Funding:** This research was funded by the Natural Sciences and Engineering Research Council of Canada (NSERC), grant number RGPIN-2023-05737, and the APC was funded by NSERC.

**Data Availability Statement:** Data can be provided based on reasonable request to the corresponding author.

**Acknowledgments:** The authors acknowledge the support of the Natural Sciences and Engineering Research Council of Canada (NSERC), Canada.

**Conflicts of Interest:** The authors declare no conflicts of interest. The funders had no role in the design of the study; in the collection, analyses, or interpretation of data; in the writing of the manuscript; or in the decision to publish the results.

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
