# Peer review of "Modeling Variability in Seismic Analysis of Concrete Gravity Dams: A Parametric Analysis of Koyna and Pine Flat Dams"

_infrastructures, doi:10.3390/infrastructures9010010_

Round 1
Reviewer 1 Report
Comments and Suggestions for Authors
The work is interesting, very well written and described and the results are correctly presented.
Model choice is essential in this kind of analysis and there is a variety of software tools and associated approaches which in principle can be used.
However, the discussion can be considered superficial and the conclusions not highly relevant. In summary, the take home message that the reader gets is that results can be highly different depending on the software and approach used. However, the readers would benefit from some more detailed discussion and some more specific recommendations as to which approach to use and which implications they have.
Other aspects may also play a relevant role in this kind of analysis, such as the mesh size or the values of the damping parameters. Please, consider mentioning this in the discussion.
At some points, results for both dams considered are compared, but the relevance of such comparison is not clear. Should results be more similar? Why? What is the origin of the differences? Do the features of the dams have implications on the choice of the model?
Overall, more emphasis should be placed on drawing practical conclusions (supported by the results) as to which approach to use.
Other comments are included in the annotated pdf file. The more relevant are those related to the conclusions, but all should be addressed.

Comments on the Quality of English LanguageSome suggestions/recommendations were included in the annotated pdf file.
Reviewer 2 Report
Comments and Suggestions for Authors
I have reviewed the manuscript entitled “Exploring modeling uncertainties in seismic analysis of dams: A case study of Koyna and Pine Flat dam” submitted to the infrastructures journal. This paper aims to compare and contrast the performance of two real-world gravity dams via three different software and various modeling assumptions. While topics related to “modeling uncertainty” are of prime importance in the field of Uncertainty Quantification, this paper couldn’t fully capture the promising features of modern FE packages. In my opinion, despite the provided quite comprehensive assessment, this paper needs a real major revision as follows:
1. In Abstract, it is stated “Addressing these uncertainties is crucial to enhance the understanding and efficacy of dam safety assessments, ensuring the resilience and longevity of aging dam in infrastructure.” However, this paper analyzes the Koyna and Pine Flat dam without capturing aging effect. Without a doubt, neglecting this important feature has a damper effect on the robustness of the results. I suggest the authors to do something in this direction.
2. Very Important: In terms of novelty, this paper is just a repetition of state-of-the-art. Comparing linear and nonlinear analysis, H and HV representation of ground motion, different dam-foundation-reservoir interaction modes are already discussed thoroughly in the literature. The only important feature can be comparing the software with various constitutive models. While high-fidelity constative models are not implemented in EAGD and ADRFS, ABAQUS can successfully address this concern. Accordingly, it is highly recommended modelling different constitutive models with ABAQUS to shed light on the synergy between “recent advancements in FE” and “modelling uncertainty” topics. Here are my suggestions (one of them is enough for publication in infrastructures):
· Moving from Lagrangian FE modelling to MMALE and even SHP frameworks;
· Aside from the classical CDP constitute models adding some new Lagrangian FE tools e.g., modified CDP with various CDP parameters;
· Adding more damage criteria, e.g., plastic strain equivalent (PEEQ). This allows for demonstrating “damage propagation” offered by ABAUQS;
· …
3. As indicated, joint opening/closing at lift joints, base sliding, nonlinear contact modelling, and heterogeneous foundation can play a vital role in modelling uncertainty quantification. In accordance with comment #2, these suggestions can boost the “novelty” ingredient of this study.
4. Some discussions are crystal clear! For example, “In contrast, Abaqus, while powerful and precise, often demands more time for analysis. Therefore, it might be reserved for critical cases or detailed assessments where computational resources are a manageable factor.”, etc. In other words, without reading this manuscript, the same conclusion can be drawn.
5. Please compare and contrast the outputs from the safety standpoint (Hint: With respect to guidelines and/or codes). To put it another way, does it really matter the deviations of outcomes? How? Why?
6. Please double-check the whole manuscript to address potential typos. For example,
-In title, “A case study of Koyna and Pine Flat dam” ---> “A case study of Koyna and Pine Flat dams”.
-In line 205, “Where” ---> “where”
-…
To recapitulate, without leveraging the power and beauty of modern FE tools, this paper cannot serve a prominent role in agreement with the aims and scope of the “Advances in Dam Engineering of the 21st Century” special issue.
Comments on the Quality of English LanguageSeems fine.
Reviewer 3 Report
Comments and Suggestions for Authors This is an interesting topic to explore. I have the following major comments for the authors to improve the manuscript: 1- the title is not consistent with the manuscript itself. First of all, the term "explore" seems strange as I am not sure how to interpret it. Does it mean you want to discuss the topic but not fully, and thus, you are exploring this topic? Secondly, the term uncertainty is not 100% consistent with the content of the manuscript. This paper mainly provides a "Parametric analysis" and not really an "uncertainty analysis". 2- Following comment #1 and also the content of the manuscript, I strongly believe the authors should adjust the Introduction and Literature Review. It is fine to keep some elements of uncertainty analysis; however, what is actually done in this manuscript is extensive parametric analysis. 3- How does Figure 2 really contribute to the body of knowledge in this manuscript? How do you use it later? Which part of your manuscript is related to this figure? 4- Does using 2-3 different software correspond to uncertainty analysis? why? What is the justification? In many companies, the results of "main" software are validated using other low-level software/code/analytical solutions. Does this mean they are also doing uncertainty analysis? This is a very critical point and also is related to comment #1 about the parametric analysis. 5- Some of the items in Figure 3 are not in the same level. For example, modeling dam-foundation interaction using 3-4 methods can be considered as variability (or if you like: uncertainty -- although I do not fully agree it is uncertainty). However, including or excluding the vertical component of the ground motion is definitely NOT an uncertain choice! You have full authority to include it in your analysis or exclude it (by the way, I don't know why someone should exclude it, considering that the vertical component is always available from seismological studies). 6- Figure 6: the cross section is definitely not the one, we typically use in seismic simulations. I see a reference to [40] however, there are hundreds of other papers showing a different cross-section. This might be the upgraded cross section after the dam is damaged. Correct? Even in that case, what is the point of showing an unusual cross-section? The old cross-section is famous because it causes failure at the neck. This new cross-section is not a state-of-practice in seismic design and there is no other dam (as far as I know) having a similar section. Therefore, whatever findings are for this cross-section, it might not be applicable to other dams. 7- I was hoping to see uncertainty in material properties as they are actually uncertain! Especially for the strange cross-section of the Koyna dam, the differences between the old and new concrete could cause stress concentration in their interface and might be interesting to study. 8- Figure 11: how do the results of modal analysis in this figure correlate to analytical solutions? for example those from Chopra? 9- Figure 12: Are we comparing apples to apples? This figure shows the dispersion in results for example for "D" and "DFR" while these two are different systems. 10- All other figures show the time history of displacement --> does it really matter to show the time history? What do we learn from comparing displacement time histories instead of only comparing their maximum values? How does showing the displacement time history contribute to our understanding of failure modes? 11- The authors should be careful with the general conclusions. For example, the first item implies the outcome comparable among three software packages for the case with the only dam. This is an intuitive conclusion. I bet even if you use SAP2000, OpenSess, and other "non-dam" software, you will get a similar conclusion. The most important conclusions should focus on dam-foundation and water interaction and also the failure mode of the concrete dams. Comments on the Quality of English Language---
Round 2
Reviewer 2 Report
Comments and Suggestions for Authors
In the previous round, I clearly raised my technical concerns about FE simulations as follows:
"-It is highly recommended to model different constitutive models with ABAQUS to shed light on the synergy between “recent advancements in FE” and “modelling uncertainty” topics. Here are my suggestions (one of them is enough for publication in infrastructures):
-..."
Given the fact that my requests are simply ignored in this round, I have no alternative but to ask for a major revision to address previous comments.
Admittedly, the study is quite comprehensive but it lacks novelty to a great extent since it is a repetition of previous studies. Please pay extra attention to the topic of the Special Issue: "Advances in Dam Engineering of the 21st Century"
Seems fine.
Round 3
Reviewer 2 Report
Comments and Suggestions for Authors
Great! Thanks for addressing my concerns. This paper can be published in its current form.
